# Rapid Overfitting of Multi-Pass SGD in Stochastic Convex Optimization

**Shira Vansover-Hager** [1]   **Tomer Koren** [1,2]   **Roi Livni** [3]

## Abstract

We study the out-of-sample performance of multi-pass stochastic gradient descent (SGD) in the fundamental stochastic convex optimization (SCO) model. While one-pass SGD is known to achieve an optimal $\Theta(1/\sqrt{n})$ excess population loss given a sample of size $n$, much less is understood about the multi-pass version of the algorithm which is widely used in practice. Somewhat surprisingly, we show that in the general non-smooth case of SCO, just a few epochs of SGD can already hurt its out-of-sample performance significantly and lead to overfitting. In particular, using a step size $\eta = \Theta(1/\sqrt{n})$, which gives the optimal rate after one pass, can lead to population loss as large as $\Omega(1)$ after just one additional pass. More generally, we show that the population loss from the second pass onward is of the order $\Theta(1/(\eta T) + \eta \sqrt{T})$, where $T$ is the total number of steps. These results reveal a certain phase-transition in the out-of-sample behavior of SGD after the first epoch, as well as a sharp separation between the rates of overfitting in the smooth and non-smooth cases of SCO. Additionally, we extend our results to with-replacement SGD, proving that the same asymptotic bounds hold after $O(n \log n)$ steps. Finally, we also prove a lower bound of $\Omega(\eta \sqrt{n})$ on the generalization gap of one-pass SGD in dimension $d = \widetilde{O}(n)$, improving on recent results of Koren et al. (2022) and Schliserman et al. (2024).

## 1. Introduction

Stochastic gradient descent (SGD) is one of the most fundamental algorithms in machine learning and optimization, widely used for training large-scale models. Theoretical analysis of SGD has traditionally focused on its behavior in the one-pass (single-epoch) setting, particularly in the context of convex optimization, where it is well understood that SGD achieves an optimal excess population loss of order $\Theta(1/\sqrt{n})$ when trained on a sample of size $n$. This rate is known to be minimax optimal under standard assumptions (Nemirovskiǐ & Yudin, 1983).

However, in practice, it is common to run multiple passes over the training data, a scheme often referred to as *multi-pass SGD*, where, in each pass, training examples are sampled without-replacement, with or without reshuffling between passes. Despite its empirical success, the theoretical implications of performing multiple passes are far less understood, particularly with respect to generalization and out-of-sample performance. The standard intuition, based on empirical observations, suggests that more training should lead to better empirical performance, yet, more steps and passes might eventually lead to overfitting as the model becomes too closely tailored to the training data being processed repeatedly.

Indeed, a substantial body of work has studied the convergence of multi-pass SGD in finite-sum problems (e.g., Recht & Re, 2012; Hardt et al., 2016; Nagaraj et al., 2019; Rajput et al., 2020; Safran & Shamir, 2020; 2021; Koren et al., 2022; Cha et al., 2023). However, these studies primarily focus on optimization convergence in terms of *empirical risk* and how it is influenced by without-replacement sampling, as opposed to with-replacement which is significantly easier to analyze theoretically. Far less attention has been given to convergence in terms of *population risk*, which quantifies out-of-sample performance and is arguably the true goal in the context of machine learning.

In fact, to the best of our knowledge, the following basic question still remains, rather surprisingly, unanswered:

> *How does the population risk performance of SGD, tuned to attain the minimax optimal rate after a single pass, deteriorate (if at all) after making just a few additional passes?*

While there are several existing upper bounds on the population risk in the multi-pass scenario (e.g., Hardt et al., 2016; Bassily et al., 2020), none of them give a meaningful, nontrivial answer to this question in the general stochas-

---

[1]Blavatnik School of Computer Science, Tel Aviv University [2]Google Research [3]School of Electrical and Computer Engineering, Tel Aviv University. Correspondence to: Shira Vansover-Hager <shirav@mail.tau.ac.il>, Tomer Koren <tkoren@tauex.tau.ac.il>, Roi Livni <rlivni@tauex.tau.ac.il>.

*Proceedings of the 42nd International Conference on Machine Learning*, Vancouver, Canada. PMLR 267, 2025. Copyright 2025 by the author(s).

tic convex setting in which the $\Theta(1/\sqrt{n})$ minimax rate of Nemirovskiĭ & Yudin (1983) applies.

Our investigation in this work reveals a rather surprising answer to this basic question: we show that even in the fundamental, well-studied convex setting, multiple passes of SGD can lead to rapid overfitting and reach a trivially-large population loss after just *one* additional pass. In particular, using the canonical stepsize of $\eta = \Theta(1/\sqrt{n})$ that leads to the minimax rate after a single pass, results in population excess risk as high as $\Omega(1)$ after merely two passes. More generally, we establish tight population excess risk bounds for multi-pass SGD using any stepsize $\eta$ and number of steps $T$. Our results cover without-replacement, single-shuffle and multi-shuffle, multi-pass SGD as well as with-replacement SGD, demonstrating similar rapid overfitting in all cases. An illustration of the population risk bounds we obtain across different epochs and stepsizes can be seen in Figure 1.

## 1.1. Our Contributions

In some more detail, we examine the out-of-sample behavior of multi-pass SGD in the fundamental setting of Stochastic Convex Optimization (SCO). In this setting, we operate with a convex and Lipschitz (yet not necessarily smooth) loss function over a bounded convex domain.

In this setting, we make the following contributions:

- We establish a tight bound of $\Theta(1/(\eta T) + \eta \sqrt{T})$ on the population excess risk of multi-pass (without replacement) SGD with stepsize $\eta$ over $T$ steps (see Theorems 3.1 and 3.3 in Section 3). This result applies to both the single-shuffle and multi-shuffle variants, but also more generally to any multi-pass scheme that processes examples according to an arbitrary sequence of permutations.

- We also prove a similar tight $\Theta(1/(\eta T) + \eta \sqrt{T})$ population excess risk bound for with-replacement SGD, that holds after $O(n \log n)$ steps (see Theorems 3.2 and 3.4).

- Finally, we also provide a new $\Omega(\eta \sqrt{n})$ lower bound on the *empirical risk* (and therefore also on the generalization gap) of single-pass SGD. Our result holds in dimension $\tilde{O}(n)$ thus improving upon previous results by Koren et al. (2022); Schliserman et al. (2024) that were established in dimension at least quadratic in $n$.

We note that all of our lower bound constructions apply in an overparameterized regime, where the dimension scales linearly in the sample size $n$. Our approach builds on recent techniques developed for analyzing the sample complexity of (approximate) empirical risk minimization in stochastic convex optimization (Feldman, 2016; Amir et al., 2021; Koren et al., 2022; Schliserman et al., 2024; Livni, 2024). In particular, we leverage methods from Livni (2024), who refined these constructions to achieve minimal dimension dependence of $O(n)$.

## 1.2. Discussion and Open Questions

It is insightful to contrast our lower bounds for multi-pass SGD with existing lower bounds for gradient methods in SCO. Most prior work on population risk lower bounds has focused on algorithms that approximately minimize the empirical risk, such as full-batch gradient descent (Amir et al., 2021; Schliserman et al., 2024; Livni, 2024). A key observation in these works is that after just a single step of gradient descent, the entire training set has been "touched" and so can be effectively memorized in the optimization iterate. Once this occurs, subsequent gradient steps can be steered toward an overfitting solution with respect to the particular training set at hand. Our results in the present paper reveal a similar memorization effect for multi-pass SGD: after a single full pass, SGD is also capable of effectively memorizing the training set and driving the optimization towards overfitting. However, a crucial (and quite remarkable) difference is that this effect cannot be manifested before completing the first pass, as doing so would contradict the classical Nemirovskiĭ & Yudin (1983) stochastic approximation upper bounds.

Our result regarding the empirical risk of one-pass SGD complement earlier work by Koren et al. (2022); Schliserman et al. (2024) and challenges the classic learning paradigm of minimizing empirical risk and generalization gap in relation to SGD. It reveals that the generalization gap is inadequate in explaining the minimax optimal generalization behavior of one-pass SGD, which constitute one of the fundamental cornerstones of convex optimization. Together with our lower bounds for multi-pass SGD, this result highlights a sharp phase transition between the first and later epochs: generalization bounds, such as those implied by algorithmic stability (Bousquet & Elisseeff, 2002; Hardt et al., 2016; Bassily et al., 2020), are ineffective during the first pass but become tight after the second pass, whereas the optimal performance in the first pass is only explained by online-to-batch (aka stochastic approximation) arguments which collapse immediately after the first pass.

**Open questions.** Our findings raise several intriguing open questions for further investigation:

- First, our construction leverages techniques for lower bounding the *generalization gap* in SCO in order to control the population risk of multi-pass SGD. However, in principle, the population risk performance of multi-pass SGD should be unrelated to its generalization gap (e.g., as in the case of one-pass SGD). In particular, it is an interesting question whether our results could be reproduced in settings where uniform convergence holds (e.g., low-norm linear predictors with a Lipschitz loss), and the generalization gap is necessarily small.

- Another interesting problem is to precisely characterize the rate at which overfitting develops during the second

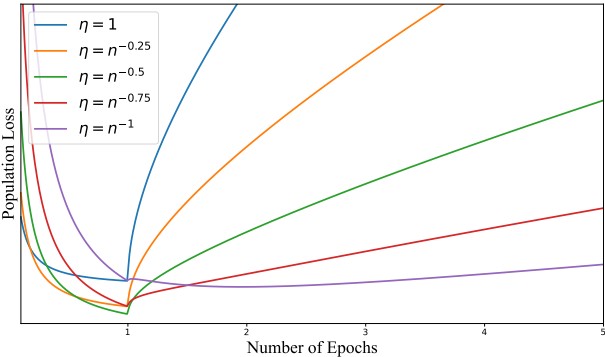

*Figure 1.* An illustration of the minmax rates for the population loss of multi-pass SGD established in Theorems 3.1 and 3.3, through $K = 5$ epochs and for different stepsizes $\eta$.

pass. Indeed, our results are only effective starting from the end of the second epoch and onward. As we discussed earlier, generalization bounds remain vacuous at the start of the second pass, and regret analysis fails due to its reliance on independent samples at each iteration. Thus, it appears that more refined techniques will be required to analyze SGD dynamics immediately after the first pass.

• Finally, our work focuses on the general SCO setting, which allows for non-smooth convex loss functions. Establishing population risk bound for multi-pass projected SGD in the analogous smooth setting remains an interesting open question that is likely to require significantly different techniques than those used in the non-smooth case. We remark that for *unconstrained* optimization, it is known that variants of smooth GD and SGD can overfit, even in dimension one (Nikolakakis et al., 2023; Zhang et al., 2023). However, these results rely on unbounded losses over unbounded domains, where learnability itself is not guaranteed.[1] In contrast, our focus is on constrained optimization—a regime where learning is possible yet ERM-based methods can fail—and the open question pertains to this case as well.

### 1.3. Additional Related Work

**Convergence bounds for multi-pass SGD.** Numerous studies have explored multi-pass SGD from an optimization perspective (e.g., Rajput et al., 2020; Safran & Shamir, 2020; Cha et al., 2023). Notably, focusing on empirical performance in finite-sum problems, Nagaraj et al. (2019) derived upper bounds for smooth functions, and Koren et al. (2022) extended these results to non-smooth func-

tions. Some works specifically study the differences in optimization convergence rates between with-replacement and without-replacement sampling (e.g., Recht & Re, 2012; Yun et al., 2021; Lai & Lim, 2020; De Sa, 2020; Safran & Shamir, 2021). Other works examine the population performance of multi-pass SGD. For example, Sekhari et al. (2021) focused on establishing upper bounds for a variant of multi-pass SGD that uses a validation set, and in a slightly non-standard formulation of SCO that allows for non-convex individual functions. In contrast, our main focus is on lower bounds for standard multi-pass SGD in the classical SCO model (Shalev-Shwartz et al., 2010). Lei et al. (2021) derive upper bounds using decreasing step sizes, without assuming Lipschitz continuity but requiring gradient norms to be bounded linearly by function values. In contrast, we establish tight bounds for fixed stepsize SGD under the standard Lipschitz assumption. Other upper bounds on population loss for the closely related with-replacement SGD can be found in Hardt et al. (2016); Lei & Ying (2020).

**Lower bounds for SGD.** In terms of lower bounds, Zhang et al. (2023) and Nikolakakis et al. (2023) analyzed SGD with replacement and multi-pass SGD, respectively, but both focused specifically on the unconstrained (smooth) case and allowed for unbounded function values, which in general, render the problem unlearnable, as discussed before. We, on the other hand, focus exclusively on bounded domains and functions values bounded by a constant, where the classical Nemirovskiĭ & Yudin (1983) results apply. More recently, Koren et al. (2022) established lower bounds for the *generalization gap* of one-pass SGD. They demonstrate that even though SGD achieves optimal population loss rates after a single pass, its empirical risk can be as high as $\Omega(\eta\sqrt{T})$ after $T$ steps. Their construction was in dimension exponential in the sample size, which was later improved to quadratic by Schliserman et al. (2024). Here, we further improve the dimensionality dependence to linear, which, as argued by Feldman (2016); Livni (2024), is optimal.

**Stochastic Convex Optimization (SCO).** Our work belongs to the study of sample complexity of learning algorithms in the fundamental SCO model. Previous research has shown that in SCO some algorithms overfit when the sample size depends on the dimension (Shalev-Shwartz et al., 2010; Feldman, 2016), while others like online-to-batch algorithms achieve dimension independent rates (Nemirovskiĭ & Yudin, 1983). This makes SCO a valuable model for testing the generalization and sample complexity of learning algorithms. Amir et al. (2021) were the first to establish a dimension dependent lower bound for the sample complexity of gradient descent (GD), they showed a population loss lower bound of $\Omega(\eta\sqrt{T})$ after $T$ steps using a step size $\eta$ in a construction with dimension exponential in the sample size $n$. Later, Schliserman et al. (2024) extended

---

[1] Indeed, with an unbounded range of function values, even evaluating the loss of a *single* model, let alone learning, becomes intractable. (E.g., without additional assumptions, standard concentration bounds scale linearly with the range of the random variables.)

this result to functions with dimensions proportional to the square of the sample size, and Livni (2024) further reduced it to linear size. This shows that to prevent GD from overfitting, a sample of size $\Omega(d)$ must be considered. While we build on ideas from these works, our focus is on SGD—rather than GD—which does generalize in its first epoch. Specifically, multi-pass SGD circumvents previous hardness constructions during its first pass and begins the second epoch with optimal generalization performance. This behavior requires a careful adaptation of existing constructions to account for the rapid deterioration of generalization after the first epoch.

**Algorithmic stability.** Our work is also related to the study of stability of learning algorithms as means to establish generalization bounds (Shalev-Shwartz et al., 2010; Bousquet & Elisseeff, 2002). The crucial advantage of stability in the multi-epoch (without replacement) setting is the fact it does not require individual examples to be sampled i.i.d.—an assumption that only holds during the first pass. Notably, Hardt et al. (2016) show bounds on uniform stability of multi-pass SGD for smooth functions and Bassily et al. (2020) show tight stability bounds for non-smooth functions. Their results demonstrate that in the non-smooth case, an additional $\Theta(\eta\sqrt{T})$ term must be added, that make such bounds vacuous (at least for the purpose of establishing risk bounds) unless the stepsize is extremely small as a function of $T$. Importantly, these bounds apply uniformly across epochs and do not explain why SGD generalizes well in its first pass and overfitting starts only from the second pass. This suggests a gap between instability and generalization behavior, which calls for a more refined analysis that goes beyond plain stability arguments.

## 2. Problem Setup

**Stochastic Convex Optimization (SCO).** We study out-of-sample performance of multi-pass SGD in the context of the fundamental stochastic convex optimization (SCO) framework. A learning problem in SCO is defined by a convex and $G$-Lipschitz loss function $f : W \times Z \to \mathbb{R}$, where $Z$ is a sample space endowed with a population distribution $\mathcal{Z}$, and $W$ is a convex, compact domain with diameter bounded by $D$. A learning algorithm receives a training set $S = \{z_1, \ldots, z_n\}$ of $n$ i.i.d. samples from $\mathcal{Z}$ and outputs a model $\widehat{w} \in W$, with the goal of minimizing the population loss:

$$F(w) = \mathbb{E}_{z \sim \mathcal{Z}}[f(w, z)].$$

Let $w^\star \in \arg\min_{w \in W} F(w)$ denote a minimizer of the objective. The empirical loss on the sample $S$ is given by:

$$F_S(w) = \frac{1}{n} \sum_{i=1}^{n} f(w, z_i),$$

and we denote by $w_S^\star \in \arg\min_{w \in W} F_S(w)$ its minimizer.

**One-pass Stochastic Gradient Descent (SGD).** One-Pass SGD receives a training set $S = \{z_1, \ldots, z_n\}$, a step size $\eta > 0$, a suffix averaging parameter $\tau \in [n + 1]$ and a first-order oracle $O_z(w) \in \partial f(w, z)$ and proceeds as:

Initialize: $\quad\quad w_0 = 0$

For $t = 1, \ldots, n$: $\quad w_{t+1} = \Pi_W \left[ w_t - \eta O_{z_t}(w_t) \right]$

Output: $\quad\quad\quad \widehat{w}_{n,\tau} = \frac{1}{\tau} \sum_{t=n-\tau+1}^{n} w_t,$

where $\Pi_W : \mathbb{R}^d \to W$ denotes the projections onto $W$. Note that we consider general $\tau$-suffix averaging as an output.

**Multi-pass SGD.** In contrast to single-pass SGD, multi-pass SGD does several passes on permutations of the same training set. It receives a training set $S = \{z_1, \ldots, z_n\}$, a number of epochs $K$, a step size $\eta > 0$, $K$ permutations $\{\pi_k : [n] \to [n]\}_{k \in [K]}$, a suffix averaging parameter $\tau \in [nK + 1]$ and a first-order oracle $O_z(w) \in \partial f(w, z)$, and proceeds as follows:

Initialize: $\quad\quad w_0 = 0$

For $t = 1, \ldots, nK$: $\quad i_t = \pi_{\lfloor t/n \rfloor}(t \bmod n)$

$\quad\quad\quad\quad\quad\quad\quad w_{t+1} = \Pi_W \left[ w_t - \eta O_{z_{i_t}}(w_t) \right]$

Output: $\quad\quad\quad \widehat{w}_{T,\tau} = \frac{1}{\tau} \sum_{t=T-\tau+1}^{T} w_t,$

where $T = nK$ is the total number of steps. We note two important special cases: the single-shuffle and multi-shuffle. In the single-shuffle case, $\pi_1$ is chosen uniformly at random, and $\pi_1 = \pi_k$ for all $k \in [K]$; in the multi-shuffle case, the $\{\pi_k\}_{k \in K}$ are chosen uniformly and independently.

**With-replacement SGD.** With replacement SGD receives a training set $S = \{z_1, \ldots, z_n\}$, a number of steps $T$, a step size $\eta > 0$, a suffix averaging parameter $\tau \in [n + 1]$ and a first-order oracle $O_z(w) \in \partial f(w, z)$. It then proceeds as follows:

Initialize: $\quad\quad w_0 = 0$

For $t = 1, \ldots, T$: $\quad i_t \sim \text{Unif}([n])$

$\quad\quad\quad\quad\quad\quad\quad w_{t+1} = \Pi_W \left[ w_t - \eta O_{z_{i_t}}(w_t) \right]$

Output: $\quad\quad\quad \widehat{w}_{n,\tau} = \frac{1}{\tau} \sum_{t=n-\tau+1}^{n} w_t.$

Note that this is not a multi-pass algorithm *per-se*, as the stochastic gradients used are independent regardless of the total number of steps $T$.

# 3. Population Risk Bounds

We present a lower bound and a matching upper bound for the population loss of multi-pass SGD for $K \leq n$ epochs.

## 3.1. Lower Bound for Multi-Pass SGD

We begin with stating the lower bound, which constitutes our main contribution:

**Theorem 3.1.** *For every $n \geq 24, 2 \leq K \leq n^2, T = nK, d = 256n, \eta > 0$ and $\tau \in [T+1]$, let $W = \{w \in \mathbb{R}^{2d+1} : \|w\|^2 \leq 1\}$ then there are a finite sample space $Z$, a distribution $\mathcal{Z}$ over $Z$, a 4-Lipschitz convex function $f(w, z)$ in $\mathbb{R}^{2d+1}$ such that for any first order oracle of $f(w, z)$ with probability $\frac{1}{2}$ if we run without replacement SGD for $T$ steps with stepsize $\eta$ and with any sequence of permutations:*

$$F(\widehat{w}_{T,\tau}) - F(w^\star) = \Omega\left( \min\left\{\eta\sqrt{T} + \frac{1}{\eta T}, 1\right\}\right).$$

We note that this result applies to single, multi shuffle and more generally for any sequence of permutations. In particular, for $\eta = \Theta(1/\sqrt{n})$ that yields minimax optimal result for one-pass, can lead up to $\Omega(1)$ population loss after just one additional pass.

We provide here a proof sketch, the full proof is presented in Section 5.

*Proof sketch.* To obtain the lower bound, we rely on the, recently introduced, notion of a *sample-dependent oracle* introduced by Livni (2024), and the reduction from sample-dependent oracles to the standard setting of stochastic convex optimization. Livni (2024) observed that if the gradients of an algorithm such as Gradient Descent (GD) are dependent on the *whole* sample, then one can utilize that to construct stochastic functions for which Gradient Descent overfits. It also turns out that one can reduce the standard setup of stochastic optimization to this weaker setting of a *sample dependent* oracle, for GD. These observations allowed the construction of distributions that cause GD to overfit at certain regimes.

SGD, though, differ from GD that observes the whole sample at the first iteration. In particular, it is impossible for the trajectory of SGD to depend on future seen examples. In turn, the reduction becomes more subtle. We note that this is not just an artifact of the proof technique, but is demonstrated by the fact that, indeed, SGD does minimize the population loss at the first epoch, and SGD circumvent the hard construction of Livni (2024). Nevertheless, we build on a similar approach, and we use a weaker sample dependent first order oracle, that cannot depend on the whole sample but may depend on past observations. As it turns out, such an oracle is enough to construct lower bounds, and also allows a reduction for the case of SGD.

Following Livni (2024), we rely on a loss function with the following structure:

$$f(w, V) = g(w, V) + \alpha h(w),$$

where $g(w, V)$ is a variant of Feldman's function (Feldman, 2016) that has the property that it has spurious empirical risk minimizers. Namely, using this function and a specific distribution, we can guarantee the high-probability existence of a "bad" vector, which for this bound will have high population loss. The second term, $h(w)$, is used to guide the iterates toward this bad vector, enabling us to achieve the lower bound. The constant $\alpha$ prevents projections from disrupting this process.

**Feldman's function and existence of a spurious empirical minimizer.** Feldman's function, $g$, relies on the existence of a set $U \subset \mathbb{R}^d$ with $|U| \geq 2^{\Omega(d)}$ such that for every $u \neq v \in U$, we have

$$\langle u, v \rangle \leq c_1 < c_2 \leq \|v\|^2.$$

We consider the following function:

$$g_1(w, V) = \max_{v \in V} \{c_1, v \cdot w\}.$$

The existence of such $U$ follows from a standard packing argument. Next, we let $Z = P(U)$, the powerset of $U$, and treat examples in $Z$ as subsets of $U$. Let $\mathcal{Z}$ be a distribution over $Z$ such that for $V \sim \mathcal{Z}$ every $u \in U$ is in $V$ with probability $\delta$.

Notice that if $w \in V$, then $g_1(w, V) \geq c_2$, and otherwise, $g_1(w, V) = c_1$. Let $S = \{V_1, \ldots, V_n\}$. For this bound we will define a bad vector as an ERM with high population loss. Notice that any vector $u_0 \notin \cup_{i=1}^n V_i$ is an ERM since $F_S(u_0) - F_S(w_S^\star) = c_1 - c_1 = 0$. If, furher, $u_0 \in U$, then its population loss is given by $F(u_0) \geq \delta(c_2 - c_1)$. By setting $\delta = \frac{1}{2}$, we can show then that any such $u_0$ is a bad ERM. By further setting $d = \Omega(n)$, we can show that with high probability such a $u_0 \in U$ exists.

**Reaching spurious empirical risk minimizer.** Feldman's function demonstrated that there exists a "bad vector" in the sense that there are minimizers of the empirical risk that yield large population loss. However, there is no guarantee that if we run SGD on Feldman's function we will reach such a minimzer (in fact, notice that SGD applied on $g$ will remain at zero). Therefore, our strategy is to add a further term $h$ that will cause SGD to converge towards $u_0$. For simplicity of the overview we assume here that $u_0 \in \{0, 1\}^d$.

For $T = O(d)$, we use the following function as $h$:

$$h_1(w) = \max_{i \in [d]} \{0, -w(i)\}.$$

Assume we ran SGD for $n$ iterations. Notice that $0 \in \partial h_1(0)$, therefore, in our setup we are allowed to return a 0 subgradient for $n$ iterations. After $n$ iterations, we start using $h$ after identifying a specific bad vector $u_0$. Here, we exploit the fact that our oracle is sample dependent, and we use $h_1$ to take one step on each of the positive coordinates of $u_0$ to reach $\eta u_0$ in at most $d$ steps.

When $T \gg d$, $h_1$ won't suffice since after $d$ such steps, we can't further increase the desired loss. To achieve the $d^3$ term, we use the following function instead:

$$h_2(w) = \max_{i \in [d]} \{0, -w(i), -(w(i) - w(i+1))\}.$$

With the following oracle:

1. If there is an index $i$ such that $w(i) = w(i+1) \geq \eta$, then output: $e_{i+1} - e_i$.

2. If there is no such index and if $i$ is the minimal index such that $w(i) = 0$, output: $-e_i$.

3. If neither of the above conditions holds, output: 0.

By applying these gradient steps only for the positive indices of $u_0$, this will allow us to achieve $\Omega(\eta\sqrt{T})$ when $T \approx d^3$.

For the case when $d \ll T \ll d^3$, we will use $h_2$, but instead of considering one index at a time, we consider blocks of indices of size $B \geq 1$ to achieve the same effect.

**Putting it together.** To compute and reach the bad vector we use the oracle's sample dependence and the fact that $0 \in \partial f(0, V)$ for all $V$, allowing us to stay at 0. During the first epoch, we remain at 0 to observe the examples. At the start of the second epoch we have memorized the entire training set $S$, we compute $u_0 \notin \cup_{i=1}^{n} V_i$ and take gradient steps to reach it using $h$.

**Reduction.** The construction we described assumes a sample-dependent oracle. The key idea behind the reduction to this case is encoding information about past examples into the iterates. This allows the gradient to use knowledge of previous examples by examining only the current iterate and sample. □

### 3.2. Lower Bound for With-Replacement SGD

Using the same technique, we also prove a lower bound for with-replacement SGD. The proof is given in Section 5 and here we give a short sketch of the proof.

**Theorem 3.2.** *For every $n \geq 24$, $2 \leq K \leq n$, $T = Kn \log n$, $d = 256n$, $\eta > 0$ and $\tau \in [T+1]$, let $W = \{w \in \mathbb{R}^{2d+1} : \|w\| \leq 1\}$ then there are a finite sample space $Z$, a distribution $\mathcal{Z}$ over $Z$, a 4-Lipschitz convex function $f(w, z)$ in $\mathbb{R}^{2d+1}$ such that for any first order oracle of $f(w, z)$ with*

*probability $\frac{1}{4}$ if we run with-replacement SGD for $T$ steps with stepsize $\eta$:*

$$F(\widehat{w}_{T,\tau}) - F(w^\star) = \Omega\left(\min\left\{\eta\sqrt{T} + \frac{1}{\eta T}, 1\right\}\right).$$

*Proof sketch.* Noticing that the key idea in the construction of Theorem 3.1 is to memorize the entire training set and then step in a "bad direction" uninterrupted, we can also apply the same approach here. In the case of with-replacement sampling, from folklore analysis of the coupon collector's problem we know that with probability $\frac{1}{2}$ we have seen the entire training set after $O(n \log n)$ steps, and from that point we can proceed similarly. □

### 3.3. Matching Upper Bounds

Finally, we give matching upper bounds to the two lower bounds presented above. Both bounds are straightforward consequences of standard techniques in convex optimization and algorithmic stability analysis; we only state the bounds here and defer proofs to Appendix A.

**Theorem 3.3.** *Let $f$ be a convex, non-smooth, and $G$-Lipschitz function, and let $S$ be a dataset of $n$ samples. If we run multi-pass SGD (either multi-shuffle or single-shuffle) with step size $\eta > 0$ for $K \geq 1$ epochs, for a total of $T = nK$ steps. Then, for any suffix average iterate $\tau = \Omega(T)$, we have the following guarantee:*

$$\mathbb{E}\left[F(\widehat{w}_{T,\tau}) - F(w^\star)\right] = O\left(\eta\sqrt{T} + \frac{1}{\eta T} + \frac{\eta T}{n}\right).$$

We will note that while the upper bound holds for the multi-shuffle and single-shuffle cases, our lower bound holds for every set of $K$ permutations. A similar result can be derived for with-replacement SGD:

**Theorem 3.4.** *Let $f$ be a convex, non-smooth, and $G$-Lipschitz function, and let $S$ be a dataset of $n$ samples. If we run with-replacement SGD with step size $\eta > 0$ for $K \geq 1$ epochs, for a total of $T = nK$ steps. Then, for any suffix average iterate $\tau = \Omega(T)$, we have the following guarantee:*

$$\mathbb{E}\left[F(\widehat{w}_{T,\tau}) - F(w^\star)\right] = O\left(\eta\sqrt{T} + \frac{1}{\eta T} + \frac{\eta T}{n}\right).$$

## 4. Empirical Risk Bounds

We further prove a lower bound of $\Omega(\eta\sqrt{T})$ for the empirical risk (equivalently, the generalization gap) of one-pass SGD using a construction in dimension $d = \widetilde{O}(n)$, improving on results from Koren et al. (2022); Schliserman et al. (2024).

**Theorem 4.1.** *For every $n \geq 17$, $d = 712n \log n$, $\eta > 0$, and $\tau \in [n+1]$ let $W = \{w \in \mathbb{R}^{2d+1} : \|w\| \leq 1\}$ then there are a datapoint set $Z$ and a distribution $\mathcal{Z}$ over $Z$, a*

*4-Lipschitz convex function $f(w, z)$ in $\mathbb{R}^{2d+1}$ such that for any first order oracle of $f(w, z)$ with probability $\frac{1}{2}$ if we run one-pass SGD with $\eta$ as a learning rate for $n$ steps then:*

$$F_S(\widehat{w}_{n,\tau}) - F_S(w_S^\star) = \Omega\left(\min\left\{\eta\sqrt{n}, 1\right\}\right).$$

The proof is deferred to Appendix C. This result strengthens the previous results that one-pass SGD cannot be fully explained by the classical learning framework of minimizing empirical loss and generalization gap. Specifically, while SGD ensures low population loss with $\eta = \Theta(1/\sqrt{n})$, it can still cause a generalization gap of up to $\Omega(1)$. Moreover, the construction is done in nearly linear dimension (up to a logarithmic factor), which is the lower bound for the dimension for such results, as uniform convergence must not hold.

*Proof sketch.* We use the same kind of construction as we did for the other proofs. Dinstinctively, though, to achieve high empirical risk, a "bad vector" is a vector that appears frequently in the training set, but is infrequent on the population. To guarantee the existence of such a vector we change the distribution $\mathcal{Z}$ so that $u_0 \in V$ with probability $\delta = O(1/n^2)$.

A similar argument (but reversed) as before demonstrates that there has to be a "bad vector" that is frequent at the first $1/16n$ of the examples, observed by the algorithm, but does not appear at the tail of the training set.

The appearance at a constant fraction of the training set ensures high empirical risk. On the other hand, absence from the tail allows uninterrupted advancement there in a similar technique as before. Our strategy then is similar to before, to compute the bad vector $u_0$, we also stay at 0, this time for $n/16$ steps. Then if $\cap_{i=1}^{n/16} V_i \neq \emptyset$, we select the minimal vector $u_0$ from this set and take gradient steps to reach it using $h$. □

## 5. Proofs of Theorems 3.1 and 3.2

We first prove the results using a weaker notion of a sample-dependent oracle and later relax this dependence. Formally, a *sample-dependent oracle* is a gradient oracle that, at each step, has access to both the current and all previous samples. We denote it as $O_S$, where $S = (S_1, \ldots, S_T)$ is the ordered sequence of sample sets. At iteration $t$, the algorithm receives $S_t$, which may be a set of samples rather than a single sample. This general setting captures a broad class of algorithms, including variants of SGD, as well as GD and batch methods. The oracle is defined as:

$$O_S(S_{1:t-1}; S_t, w_t) = \frac{1}{|S_t|} \sum_{z \in S_t} O_z(S_{1:t-1}; w_t),$$

when $S_{1:0} = \emptyset$, $S_{1:t-1} = (S_1, \ldots, S_{t-1})$ and $O_z(S_{1:t-1}; w) \in \partial f(w, z)$. The trajectory induced by $O_S$ which is initialized at $w_0 = 0$ is specified by the following equation:

$$w_{t+1} = w_t - \eta O_S(S_{1:t-1}; w_t, S_t).$$

We will state the following general Lemma for an upper bound of $\Omega(\eta\sqrt{T})$ and later show how it proves Theorems 3.1 and 3.2.

**Lemma 5.1.** *For every $n$, $\tau_{epoch} \geq 24$, $K \geq 2$, and step size $\eta > 0$, define $T = \tau_{epoch}K$ and $d = 256n$ and let $W = \{w \in \mathbb{R}^{2d} : \|w\| \leq 1\}$. There exist a finite dataset $Z$, a distribution $\mathcal{Z}$ over $Z$, a 3-Lipschitz convex function $f(w, z)$ in $\mathbb{R}^{2d}$, as well as a sample-dependent first-order oracle $O_S$ such that for every training set $S \sim \mathcal{Z}^n$ drawn i.i.d. and every sequence of samples $S = (S_1, \ldots, S_T)$ where $S_t \subset S$ for all $t \in [T]$, if with probability $p$ it holds that $\bigcup_{t=1}^{\tau_{epoch}} S_t = S$, then with probability at least $\frac{1}{2}p$, for every suffix averaging $\tau$:*

$$F(\widehat{w}_{T,\tau}) - F(0) = \Omega\left(\min\left\{\eta\sqrt{\min\{n^3, T\}}, 1\right\}\right).$$

This Lemma shows that once an algorithm memorizes the entire training set we can quickly lead it to overfit.

*Proof.* We will prove here the Lemma for the case $K \geq 34$. For $2 \leq K \leq 34$ the proof is deferred to Appendix B. First we notice that there exists $d \leq d' < 2d$ such that $d' = 2^m$ for some $m \in \mathbb{N}$. Our construction will be over $\mathbb{R}^{d'}$ which of course implies that such a construction can be done in $\mathbb{R}^{2d}$ using only a subspace of dimension $d'$. From the proof of Lemma 8 in Livni (2024) there exists $U \subset \{0, 1\}^{d'}$ such that:

$$\forall u \neq v \in U, u \cdot v \leq \frac{5}{16}d' \quad \text{and} \quad \|u\|^2 = \frac{7}{16}d',$$

and

$$|U| > e^{d'/258} > e^{d/258} \geq 2^{d/256} = 2^n.$$

We will let $Z = P(U)$ and define a distribution $\mathcal{Z}$ over $Z$ such that for a random sample $V \sim \mathcal{Z}$, each $u \in U$ lies in $V$ with probability $\frac{1}{2}$. Let $\alpha = \min\left\{\frac{1}{\eta\sqrt{2T}}, 1\right\}$. First we will assume $T \leq d'^3$ and we will consider the case where $T > d'^3$ at the end. Let $B \in \mathbb{N}$ be such that:

$$d' \leq B\left(\frac{T}{34}\right)^{1/3} < 2d'.$$

One can show that without loss of generality we can assume $B$ is also a power of 2, in particular $d'$ is divisible by $B$ for a large enough $T$. This will imply two useful facts:

$$\frac{d'^3}{B^3} \leq \frac{T}{34} \implies \tau_{epoch} + \frac{d'^3}{B^3} \leq \frac{T}{34} + \frac{T}{34} = \frac{1}{17}T, \quad (1)$$

since $T \geq 34\tau_{\text{epoch}}$ and

$$T \leq \frac{272 d'^3}{B^3}, \tag{2}$$

which we will use later. We consider blocks of indices so we will present the following notation: for two sets of indices $I, J \subset [d']$ denote $I \prec J$ if $\max\{i \in I\} < \min\{j \in J\}$ and denote:

$$e_I = \frac{1}{\sqrt{|I|}} \sum_{i \in I} e_i \quad \text{and} \quad w(I) = \frac{1}{\sqrt{|I|}} \sum_{i \in I} w(i).$$

We will define the functions:

$$h(w) = \max \left\{ 0, \max_{I \subset [d'], |I| = B} \{-w(I)\}, \right.$$

$$\left. \max_{I \prec J \subset [d'], |I| = |J| = B} \{-(w(I) - w(J))\} \right\}$$

$$g(w, V) = \frac{1}{\sqrt{d'}} \max_{v \in V} \left\{ \frac{45 \eta \alpha d'^2}{2 \cdot 16^2 B^{1.5}}, w \cdot v \right\}.$$

Finally our loss function will be:

$$f(w, V) = g(w, V) + \alpha h(w). \tag{3}$$

Notice that $f$ is convex and 3-Lipschitz. Next we define a sample dependent oracle $O_S$. Given $w$ and all examples seen so far $S_{1:t} = \{S_1, \ldots, S_t\}$:

1. If $|S_{1:t}| \leq \tau_{\text{epoch}}$ output 0.

2. Otherwise check in lexicographic order if there exists $u_0 \in U$ such that $u_0 \notin (\cup_{t'=1}^{\tau_{\text{epoch}}} \cup_{V \in S_{t'}} V)$.
   - If there doesn't exist such $u_0$ or $u_0 \in (\cup_{t'=\tau_{\text{epoch}}+1}^{t} \cup_{V \in S_{t'}} V)$ output an arbitrary sub-gradient.
   - Otherwise we will compute the $\frac{7d'}{16}$ indices of $u_0$ that hold $u_0 = 1$ denote them $J = \{j_1, \ldots, j_{7d'/16}\}$. Such a set exists since $\|u_0\|^2 = \frac{7d'}{16}$ and $u_0 \in \{0, 1\}^{d'}$. We will divide the indices in $J$ to blocks of size $B$, for $m = 1, \ldots, \frac{7d'}{16B}$:

     $$I_m = \{j_{(m-1) \cdot B + 1}, \ldots, j_{m \cdot B}\}.$$

   We have three scenarios:
   (a) If there is a block $I_j$ such that $w(I_j) = w(I_{j+1}) > 0$ then output: $\alpha(e_{I_{j+1}} - e_{I_j})$.
   (b) If there is no such block and if $I_j$ is the minimal block such that $w(I_j) = 0$ output: $-\alpha e_{I_j}$.
   (c) If both of the conditions stated above do not hold output: 0.

We will denote the following event:

$$\mathcal{E} = \left\{ \cup_{t=1}^{\tau_{\text{epoch}}} S_t = S \text{ and } \exists u_0 \in U : u_0 \notin \cup_{t=1}^{\tau_{\text{epoch}}} \cup_{V \in S_t} V \right\}. \tag{4}$$

We will prove $O_S$ is a valid oracle in the following lemma whose proof is deferred to Appendix B.1.

**Lemma 5.2.** *$O_S$ as stated above is a valid sample dependent first order oracle of $f$ as defined in Equation (3). Furthermore if $\mathcal{E}$ holds it will induce a trajectory such that we never leave the unit ball and no projection takes place.*

We will now assume that $\mathcal{E}$ holds. Since $|U| \geq 2^n$ from Lemma E.2 and from the definition of $\tau_{\text{epoch}}$,

$$\Pr[\mathcal{E}] = \Pr[\cup_{t=1}^{\tau_{\text{epoch}}} S_t = S] \cdot \Pr[\exists u_0 \in U : u_0 \notin \cup_{V \in S} V]$$

$$\geq \frac{1}{2} p.$$

Next we will assume that $\mathcal{E}$ holds. Then after at most $T' = \tau_{\text{epoch}} + 1 + \sum_{t=1}^{\frac{7d'}{16B}} \sum_{t'=1}^{t} t'$ steps we will have that for every $i \in I_t$, $w_{T'}(i) = \alpha \eta \sqrt{B} \left( \frac{7d'}{16B} + 1 - t \right)$, and for every $t' \geq T'$, $w_{t'} = w_{T'}$ which implies:

$$w_{t'} \cdot u_0 = w_{T'} \cdot u_0 = \sqrt{B} \eta \, \alpha \sum_{t=1}^{\frac{7d'}{16B}} \left( \frac{7d'}{16B} + 1 - t \right)$$

$$= \sqrt{B} \eta \alpha \sum_{t=1}^{\frac{7d'}{16B}} t$$

$$\geq \frac{\sqrt{B} \eta \alpha}{2} \cdot \left( \frac{7d'}{16B} \right)^2$$

$$= \sqrt{B} \eta \alpha \frac{49 d'^2}{2 \cdot (16B)^2}.$$

Since $T' \leq \tau_{\text{epoch}} + \frac{d'^3}{B^3} \leq \frac{1}{17} T$ (see Equation (1)), for every suffix averaging $\tau$:

$$\widehat{w}_{T,\tau} \cdot u_0 \geq \frac{16}{17} w_{T'} \cdot u_0 \geq \sqrt{B} \eta \alpha \frac{46 d'^2}{2 \cdot (16B)^2}.$$

Also because we assumed $T \leq \frac{272 d'^3}{B^3}$ (see Equation (2)), and with probability at least $\frac{1}{2}$ $u_0$ will appear in a fresh sample:

$$F(\widehat{w}_{T,\tau}) - F(0) \geq \frac{1}{2} \left( \frac{46 \sqrt{B} \alpha \eta d'^{1.5}}{2 \cdot (16B)^2} - \frac{45 \sqrt{B} \alpha \eta d'^{1.5}}{2 \cdot (16B)^2} \right)$$

$$\geq \frac{\alpha \eta}{4 \cdot 16^2} \left( \frac{d'}{B} \right)^{1.5}$$

$$\geq \frac{1}{4 \cdot \sqrt{2} \cdot \sqrt{272} \cdot 16^2} \min\{\eta \sqrt{T}, 1\}.$$

This concludes the proof for the case $T \leq d'^3$, when $T > d'^3$ if we take the construction with $T = d'^3$ then after $T$ steps our oracle will keep giving 0, hence we can use the above construction for any $T' > T$. So for $T > d'^3$ we have with probability $\frac{1}{2} p$:

$$F(\widehat{w}_{T,\tau}) - F(0) \geq \frac{1}{4\sqrt{2} \cdot \sqrt{272} \cdot 16^2} \min\{\eta \sqrt{d'^3}, 1\}$$

$$\geq \frac{1}{4\sqrt{2} \cdot \sqrt{272} \cdot 16^2} \min\{\eta \sqrt{(256n)^3}, 1\}.$$

Overall in both cases with probability $\frac{1}{2}p$:

$$F(\widehat{w}_{T,\tau}) - F(0) = \Omega\left(\min\left\{1, \eta\sqrt{\min\{n^3, T\}}\right\}\right). \quad \square$$

We will now show how the lemma gives the desired results.

*Proof of Theorem 3.1.* The $\Omega(1/(\eta T) + \eta)$ term is given in Lemma E.1. For the $\Omega(\eta\sqrt{T})$ term we will use Lemma 5.1. Let $S = (\{z_{i_1}\}, \dots, \{z_{i_T}\})$ be the ordered sequence of samples given to multi-pass SGD during a run of $T$ steps with training set $S$ of $n$ samples. After the first epoch we are sure to have observed the entire training set, so for $\tau_{\text{epoch}} = n$ and $p = 1$ from Lemma 5.1 we get the result using a sample-dependent oracle. We can relax this dependence using the reduction in Lemma D.2 and conclude the proof. $\quad \square$

*Proof of Theorem 3.2.* The $\Omega(1/(\eta T) + \eta)$ term is given in Lemma E.1. For the $\Omega(\eta\sqrt{T})$ term we will use Lemma 5.1. Let $S = (\{z_{i_1}\}, \dots, \{z_{i_T}\})$ be the ordered sequence of samples given to with-replacement SGD during a run of $T$ steps with training set $S$ with $n$ samples. From folklore analysis of the coupon collector's problem it is known that with probability $\frac{1}{2}$ after at most $n \log n$ iterations we have memorized the entire training set, so for $\tau_{\text{epoch}} = n \log n$ and $p = \frac{1}{2}$ from Lemma 5.1 we get the result using a sample-dependent oracle. We can relax this dependence using the reduction in Lemma D.2 and conclude the proof. $\quad \square$

## Acknowledgments

This project has received funding from the European Research Council (ERC) under the European Union's Horizon 2020 research and innovation program (grant agreements OPTGEN-101078075 and FoG-101116258). Views and opinions expressed are however those of the author(s) only and do not necessarily reflect those of the European Union or the European Research Council. Neither the European Union nor the granting authority can be held responsible for them. This work received additional support from the Israel Science Foundation (ISF; grants numbers 3174/23 and 2188/20), and a grant from the Tel Aviv University Center for AI and Data Science (TAD) and a grant from the Israeli Council of Higher Education.

## Impact Statement

This paper presents work whose goal is to advance the field of Machine Learning. There are many potential societal consequences of our work, none which we feel must be specifically highlighted here.

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

# A. Proofs of Upper Bounds

To establish the upper bounds, we will use stability arguments, and state the following definition for uniform stability:

**Definition A.1.** A randomized algorithm $A$ is $\epsilon$-uniformly stable if for all data sets $S, S' \in Z^n$ such that $S$ and $S'$ differ in at most one example, we have:

$$\sup_z \mathbb{E}\left[f(A(S), z) - f(A(S), z')\right] \leq \epsilon.$$

We will denote by $\epsilon_{\text{stab}}$ the infimum over such $\epsilon$.

We recall the important Lemma stating that uniform stability implies generalization in expectation:

**Lemma A.2** (Lemma 7 in Bousquet & Elisseeff (2002)). *Let $A$ be an algorithm that is $\epsilon$ uniformly stable, then:*

$$\left|\mathbb{E}_{S,A}\left[F(A(S)) - F_S(A(S))\right]\right| \leq \epsilon.$$

*In particular using our notation:*

$$\left|\mathbb{E}_{S,A}\left[F(A(S)) - F_S(A(S))\right]\right| \leq \epsilon_{stab}.$$

Our upper bounds will be established using the following standard inequality:

$$\mathbb{E}\left[F(\widehat{w}) - F(w^\star)\right] = \mathbb{E}\left[F(\widehat{w}) - F_S(\widehat{w})\right] + \mathbb{E}\left[F_S(\widehat{w}) - F_S(w_S^\star)\right] + \overbrace{\mathbb{E}\left[F_S(w_S^\star) - F_S(w^\star)\right]}^{\leq 0} + \overbrace{\mathbb{E}\left[F_S(w^\star) - F(w^\star)\right]}^{=0}$$
$$\leq \mathbb{E}\left[F(\widehat{w}) - F_S(\widehat{w})\right] + \mathbb{E}\left[F_S(\widehat{w}) - F_S(w_S^\star)\right]$$

Using Lemma A.2 yeilds:

$$\mathbb{E}\left[F(\widehat{w}) - F(w^\star)\right] \leq \epsilon_{\text{stab}} + \mathbb{E}\left[F_S(\widehat{w}) - F_S(w_S^\star)\right] \tag{5}$$

We will now continue to prove the upper bounds.

*Proof of Theorem 3.3.* We will prove the theorem for the average of the iterates. The proof for any suffix average $\tau = \Omega(T)$ can be derived using similar arguments. To bound the stability we will use the following Lemma which was originally given for multi-pass SGD with single-shuffle but can easily be extended to the multi shuffle case using similar arguments.

**Lemma A.3** (Theorem 3.4 in Bassily et al. (2020)). *Let $f : W \times Z \to \mathbb{R}$ be a convex $G$-Lipschitz function. Then the uniform stability of multi-pass SGD with multi-shuffling or single-shuffling is bounded as:*

$$\epsilon_{stab} \leq 2G^2 \eta \sqrt{T} + 4G^2 \frac{\eta T}{n}.$$

To bound the optimization error we will use the following:

**Lemma A.4** (Theorem 6 in Koren et al., 2022). *Let $f : W \times Z \to \mathbb{R}$ be a convex and $G$-Lipschitz function. And let $S = \{z_1, \ldots, z_n\}$ be some training set. Consider running $K \geq 1$ epochs of without replacement SGD over $f$ and $S$. Then, we have the following guarantee for $\widehat{w} = \frac{1}{nK} \sum_{t=1}^{T+1} w_t$:*

- *For the multi-shuffle case:*

$$\mathbb{E}\left[F_S(\widehat{w}) - F_S(w_S^\star)\right] \leq 2\eta G^2 \sqrt{n} + \frac{D^2}{2\eta nK} + \frac{1}{2}\eta G^2.$$

- *For the single-shuffle case:*

$$\mathbb{E}\left[F_S(\widehat{w}) - F_S(w_S^\star)\right] \leq 8G^2 \eta \sqrt{nK} + 8G^2 \eta K + \frac{D^2}{2\eta nK} + \frac{\eta G^2}{2}.$$

So for both cases

$$\mathbb{E}\left[F_S(\widehat{w}) - F_S(w_S^\star)\right] \le 8G^2\eta\sqrt{nK} + 8G^2\eta K + \frac{D^2}{2\eta nK} + \frac{\eta G^2}{2}.$$

Using Equation (5):

$$\mathbb{E}\left[F(\widehat{w}) - F(w^\star)\right] \le \epsilon_{\text{stab}} + \mathbb{E}\left[F_S(\widehat{w}) - F_S(w_S^\star)\right]$$

$$\le 2G^2\left(\eta\sqrt{nK} + 2\eta K\right) + 8G^2\eta\sqrt{nK} + 8G^2\eta K + \frac{D^2}{2\eta nK} + \frac{\eta G^2}{2}$$

$$= \frac{D^2}{2nK\eta} + 10G^2\eta\sqrt{nK} + 12G^2\eta K + \frac{1}{2}G^2\eta.$$

So we have:

$$\mathbb{E}\left[F(\widehat{w}) - F(w^\star)\right] = O\left(\frac{1}{nK\eta} + \eta\sqrt{nK} + \eta K\right) = O\left(\frac{1}{\eta T} + \eta\sqrt{T} + \frac{\eta T}{n}\right).$$

this concludes the proof. □

*Proof of Theorem 3.4.* We will prove the theorem for the average of the iterates. The proof for any suffix average $\tau = \Omega(T)$ can be derived using similar arguments. To bound the stability we will use the following Lemma

**Lemma A.5** (Theorem 3.3 in Bassily et al. (2020)). *Let $f : W \times Z \to \mathbb{R}$ be a convex G-Lipschitz function. Then the uniform stability of SGD with-replacement is bounded as:*

$$\epsilon_{stab} \le 4G^2\eta\sqrt{T} + 4G^2\frac{\eta T}{n}.$$

To bound the optimization error we will use the following classical result from Nemirovskiĭ & Yudin (1983).

**Lemma A.6.** *Assume we run SGD with stepsize $\eta > 0$ on a convex function $F' = \mathbb{E}_Z[f'(w, z)]$, for some distribution $Z$, assume further that $f'$ is G Lipschitz and $\|w_0 - w^\star\| \le D$. Let $\widehat{w}$ denote the average of the T iterates of the algorithm. Then we have:*

$$F'(\widehat{w}) - F'(w^\star) \le \frac{D^2}{2\eta T} + \frac{1}{2}G^2\eta.$$

Applying Lemma A.6 on the empirical loss $F' = F_S$ with $Z$ being the uniform distribution over $S$ we have:

$$\mathbb{E}\left[F_S(\widehat{w}) - F_S(w_S^\star)\right] \le \frac{D^2}{2nK\eta} + \frac{1}{2}G^2\eta.$$

Using Equation (5) and putting everything together:

$$\mathbb{E}\left[F(\widehat{w}) - F(w^\star)\right] \le \epsilon_{\text{stab}} + \mathbb{E}\left[F_S(\widehat{w}) - F_S(w_S^\star)\right]$$

$$\le \frac{D^2}{2nK\eta} + \frac{1}{2}G^2\eta + 4G^2\left(\eta\sqrt{nK} + \eta K\right)$$

$$\le \frac{D^2}{2nK} \cdot \frac{1}{\eta} + 2G^2\eta(\sqrt{nK} + K).$$

So we have:

$$\mathbb{E}\left[F(\widehat{w}) - F(w^\star)\right] = O\left(\frac{1}{nK\eta} + \eta\sqrt{nK} + \eta K\right) = O\left(\frac{1}{\eta T} + \eta\sqrt{T} + \frac{\eta T}{n}\right).$$

this concludes the proof. □

## B. Proof of Lemma 5.1

We will prove now the Lemma for $2 \leq K \leq 34$. The proof for the $K \geq 34$ case is given in Section 5. From the proof of Lemma 8 in Livni (2024) there exists $U \subset \{0, 1\}^d$ such that for every $u \neq v \in U$:

$$\langle u, v \rangle \leq \frac{5d}{16} \leq \frac{7d}{16} = \|v\|^2$$

and,

$$|U| \geq e^{d/258} \geq 2^n.$$

We will take the power set of $U$ as the sample space: $Z = P(U)$, identifying samples as subsets of $U$. Define a distribution $\mathcal{Z}$ over $Z$ such that for a random sample $V \sim \mathcal{Z}$, each $u \in U$ lies in $V$ with probability $\frac{1}{2}$. Let $\alpha = \min\left\{1, \frac{1}{\eta\sqrt{T}}\right\}$, denote $w^{(1)} = w[1 : d]$ and $w^{(2)} = w[d + 1 : 2d]$. We will introduce the following notation for a block of indices $I \subset [d]$:

$$e_I = \frac{1}{\sqrt{|I|}} \sum_{i \in I} e_i \text{ and } w(I) = \frac{1}{\sqrt{|I|}} \sum_{i \in I} w(i).$$

For $B = \frac{3d}{\tau_{\text{epoch}}}$ consider the following functions:

$$g(w, V) = \frac{1}{\sqrt{d}} \max_{v \in V} \left\{ \frac{5\alpha}{16\sqrt{B}} \eta d, \left( w^{(1)} + w^{(2)} + \frac{5\eta\alpha}{7\sqrt{B}} \vec{1} \right) \cdot v \right\}, \tag{6}$$

$$h(w) = \frac{5}{7} \cdot \max_{I \subset [d]:|I| \leq B} \left\{ 0, w^{(1)}(I) \right\} + \frac{2}{7} \cdot \max_{I \subset [d]:|I| \leq B} \left\{ 0, -w^{(2)}(I) \right\}. \tag{7}$$

Finally, our loss function will be:

$$f(w, V) = g(w, V) + \alpha h(w). \tag{8}$$

It holds that $f$ is convex and 3-Lipschitz. Next we define a sample dependent oracle $O_S$. Given $w_t$ and all the examples seen so far $S_{1:t} = \{S_1, \ldots, S_t\}$:

1. If $|S_{1:t}| \leq \tau_{\text{epoch}}$ output 0.

2. Otherwise we can check if there exists $u_0 \in U$ such that $u_0 \notin (\bigcup_{t'=1}^{\tau_{\text{epoch}}} \bigcup_{V \in S_{t'}} V)$. We will check this in lexicographic order.

   (a) If there doesn't exist such $u_0$, or $u_0 \in (\bigcup_{t'=\tau_{\text{epoch}}+1}^{t} \bigcup_{V \in S_{t'}} V)$, output an arbitrary sub-gradient.

   (b) Otherwise we compute the following set:

   $$J_t^0 = \left\{ i \in [d] : u_0(i) = 0 \text{ and } \left[ w_t^{(1)} + w_t^{(2)} + \frac{5\eta\alpha}{7\sqrt{B}} \vec{1} \right](i) > 0 \right\}$$

   if $|J_t^0| > 0$ choose $\min\{B, |J_t^0|\}$ indices out of $J_t^0$ denote them $Z_t$. If $|Z_t| < |J_t^0|$ let $O_t = \emptyset$. Otherwise we will let $O_t$ be at most $B$ elements of the following set:

   $$J_t^1 = \left\{ 2i : i \in [d] \text{ and } u_0(i) = 1 \text{ and } \left[ w_t^{(1)} + w_t^{(2)} + \frac{5\eta\alpha}{7\sqrt{B}} \vec{1} \right](i) < \frac{\eta\alpha}{\sqrt{B}} \right\}.$$

   Finally, output

   $$\alpha \left( \frac{5}{7} e(Z_t) - \frac{2}{7} e(O_t) \right).$$

We will denote the following event:

$$\mathcal{E} = \left\{ \cup_{t=1}^{\tau_{\text{epoch}}} S_t = S \quad \text{and} \quad \exists u_0 \in U : u_0 \notin (\cup_{t=1}^{\tau_{\text{epoch}}} \cup_{V \in S_t} V) \right\} \tag{9}$$

We will prove $O_S$ is a valid oracle and that under the event of $\mathcal{E}$ no projections take place in the following Lemma whose proof is deferred to the end of the proof.

**Lemma B.1.** $O_S$ *stated above is a valid sample dependent first order oracle of $f$ defined in Equation* (8). *Furthermore it will induce a trajectory such that if $\mathcal{E}$ holds then we never leave the unit ball and no projections take place.*

Since $|U| \geq 2^n$ from Lemma E.2 and the definition of $\tau_{\text{epoch}}$:

$$\Pr[\mathcal{E}] = \Pr[\cup_{t=1}^{\tau_{\text{epoch}}} S_t = S] \cdot \Pr[\exists u_0 \in U : u_0 \notin (\cup_{t=1}^{\tau_{\text{epoch}}} \cup_{V \in S_t} V)| \cup_{t=1}^{\tau_{\text{epoch}}} S_t = S]$$

$$= \Pr[\cup_{t=1}^{\tau_{\text{epoch}}} S_t = S] \cdot \Pr[\exists u_0 \in U : u_0 \notin \cup_{V \in S} V] \geq \frac{1}{2}p$$

Next we will assume that $\mathcal{E}$ holds. According to $O_S$ in every step we change at least $B$ indices unless we output 0. So after at most $T' = \tau_{\text{epoch}} + \lceil \frac{d}{B} \rceil$ steps we will have $|J_t^0| = |J_t^1| = 0$ meaning that $\left[ w_{T'}^{(1)} + w_{T'}^{(2)} + \frac{5\eta\alpha}{7\sqrt{B}} \vec{1} \right] = \frac{\eta\alpha}{\sqrt{B}} u_0$. Note that:

$$T' = \tau_{\text{epoch}} + \lceil \frac{d}{B} \rceil$$

$$\leq \tau_{\text{epoch}} + \lceil \frac{\tau_{\text{epoch}}}{3} \rceil$$

$$\leq \tau_{\text{epoch}} + \frac{\tau_{\text{epoch}}}{3} + 1$$

$$\leq \tau_{\text{epoch}} + \frac{\tau_{\text{epoch}}}{3} + \frac{\tau_{\text{epoch}}}{24} \qquad\qquad \tau_{\text{epoch}} \geq 24$$

$$\leq \frac{3\tau_{\text{epoch}}}{2}.$$

From the way we defined $O_S$ for every $t' \geq T'$: $w_{t'} = w_{T'}$. For every $i \in [d]$ such that $u_0(i) = 1$:

$$\left[ \widehat{w}_{T,\tau}^{(1)} + \widehat{w}_{T,\tau}^{(2)} + \frac{5\eta\alpha}{7\sqrt{B}} \vec{1} \right](i) = \frac{1}{\tau} \sum_{t=T-\tau+1}^{T} \left[ w_t^{(1)} + w_t^{(2)} + \frac{5\eta\alpha}{7\sqrt{B}} \vec{1} \right](i)$$

$$\geq \frac{1}{T} \sum_{t=1}^{T} \left[ w_t^{(1)} + w_t^{(2)} + \frac{5\eta\alpha}{7\sqrt{B}} \vec{1} \right](i)$$

$$= \frac{\eta\alpha}{T\sqrt{B}} \left( \frac{5}{7} \cdot T' + (T - T') \right)$$

$$\geq \frac{\eta\alpha}{\sqrt{B}} \left( \frac{5}{7} \cdot \frac{3}{2} + \frac{1}{2} \right) = \frac{11\eta\alpha}{14\sqrt{B}}$$

Which implies:

$$\left[ \widehat{w}_{T,\tau}^{(1)} + \widehat{w}_{T,\tau}^{(2)} + \frac{5\eta\alpha}{7\sqrt{B}} \vec{1} \right] \cdot u_0 = \frac{7d}{16} \cdot \frac{11\eta\alpha}{14\sqrt{B}} = \frac{11\eta\alpha d}{32\sqrt{B}}$$

From the definition of $\mathcal{Z}$, with probability $\frac{1}{2}$, $u_0$ will appear in a new random sample so for every suffix averaging $\tau$:

$$F(w_{T,\tau}) - F(0) \geq \frac{1}{2} \left( \frac{1}{\sqrt{d}} \left[ \widehat{w}_{T,\tau}^{(1)} + \widehat{w}_{T,\tau}^{(2)} + \frac{5\eta\alpha}{7\sqrt{B}} \vec{1} \right] \cdot u_0 - \frac{5\alpha\eta}{16 \cdot \sqrt{B}} \sqrt{d} \right)$$

$$= \frac{1}{2} \left( \frac{11\eta\alpha\sqrt{d}}{32\sqrt{B}} - \frac{5\alpha\eta}{16 \cdot 2\sqrt{c}} \sqrt{d} \right)$$

$$= \frac{\alpha\eta\sqrt{d}}{2 \cdot 32 \cdot \sqrt{B}}.$$

Overall we have shown that with probability $\frac{1}{2}$ for every suffix averaging $\tau$:

$$F(\widehat{w}_{T,\tau}) - F(0) \geq \frac{\alpha\eta}{2 \cdot 32} \cdot \sqrt{\frac{d}{B}} = \frac{\alpha\eta}{2 \cdot 32} \cdot \sqrt{\frac{\tau_{\text{epoch}}}{3}} = \Omega\left( \min\left\{ 1, \eta\sqrt{T} \right\} \right),$$

since $T = O(\tau_{\text{epoch}})$, the proof is complete. $\qquad\qquad\qquad\qquad\qquad\qquad\qquad\qquad\qquad\qquad\qquad\square$

## B.1. Proofs for Auxiliary Lemmas

We will prove here Lemmas 5.2 and B.1 that concern gradient oracles that were given in the proofs of Lemma 5.1.

*Proof of Lemma B.1.* First we note that for every $u \in U$ :

$$\frac{5\eta\alpha}{7\sqrt{B}} \vec{1} \cdot v = \frac{7}{16} d \cdot \frac{5\eta\alpha}{7\sqrt{B}} = \frac{5\eta\alpha d}{16\sqrt{B}} \tag{10}$$

So $0 \in \partial f(0, V)$ for every $V \in P(U)$. For that reason we can stay at 0 for as long as we want. Note that it suffices to show that if $\mathcal{E}$ holds the oracle is valid since for every other scenario we output either 0 or an arbitrary subgradient. If $\mathcal{E}$ holds we are clearly outputting a subgradient of $\alpha h(w)$ in Item 2b, so it is left to prove that for every $V \in S$ we have that $0 \in \partial g(w_t, V)$. Indeed,

- If for all $i \in [d]$: $\left[ w_t^{(1)} + w_t^{(2)} + \frac{5\eta\alpha}{7\sqrt{B}} \vec{1} \right](i) \leq \frac{5\eta\alpha}{7\sqrt{B}}$ then as we saw in Equation (10) this holds.

- If there exists $i \in [d]$ such that: $\left[ w_t^{(1)} + w_t^{(2)} + \frac{5\eta\alpha}{7\sqrt{B}} \vec{1} \right](i) > \frac{5\eta\alpha}{7\sqrt{B}}$ this can happen only after we have zeroed all the entries that $u_0$ has zeros on. So we are left only with positive coordinates of $u_0$ and the proof is completed by noticing that since $u_0 \notin \cup_{V \in S} V$, for every $v \in \cup_{V \in S} V$:

$$\left[ w_t^{(1)} + w_t^{(2)} + \frac{5\eta\alpha}{7\sqrt{B}} \vec{1} \right] \cdot v \leq \frac{\eta\alpha}{\sqrt{B}} u_0 \cdot v = \frac{5\eta\alpha d}{16\sqrt{B}}.$$

This completes the proof that $O_S$ is valid. To see that projections don't take place according to this oracle for every $t \in [T]$ and $i \in [2d]$ we have that: $w_t(i) \leq \frac{\eta\alpha}{\sqrt{B}}$ which implies:

$$\|w_t\|^2 \leq \eta\alpha \cdot \sqrt{\frac{2d}{B}} \leq \eta\alpha \sqrt{\frac{2\tau_{\text{epoch}}}{3}} \leq \eta\alpha\sqrt{T} \leq 1,$$

since $\alpha = \min\left\{1, \frac{1}{\eta\sqrt{T}}\right\}$. $\qquad\square$

*Proof of Lemma 5.2.* For the first part of the oracle note that $0 \in \partial f(0, V)$ so we can stay at 0 for as long as we like, in particular for $\tau_{\text{epoch}}$ steps. For the second part of the oracle, if there doesn't exists such $u_0$ or $u_0 \in (\bigcup_{t'=\tau_{\text{epoch}}+1}^{t} \bigcup_{V \in S_{t'}} V)$ we output a valid sub-gradient by definition. Otherwise, we are clearly taking gradient steps for $\alpha h(w)$. It is left to show that in this case $0 \in \partial g(w_t, V)$. Indeed, this event ensures that up to this point the oracle only output 0 or executes Items 2a to 2c, so our only nonzero coordinates are positive coordinates of $u_0$. Also, for every $v \in (\bigcup_{V \in S_t} V)$, we have that $v \neq u_0$ so $v \cdot u_0 \leq \frac{5d'}{16}$ which implies:

$$w_t \cdot v \leq \sum_{i=1}^{d'} w_t(i) \mathbb{1}\{v(i) = u_0(i) = 1\} \leq \sum_{i=1}^{\frac{5d'}{16B}} \sqrt{B}\eta\alpha \left( \frac{7d'}{16B} + 1 - t \right) \leq \sum_{i=0}^{\frac{5d'}{16B}} \sqrt{B}\eta\alpha \left( \frac{7d'}{16B} - t \right) \leq$$

$$\leq \sqrt{B}\eta\alpha \left( \frac{5d'}{16B} \cdot \frac{7d'}{16B} - \frac{1}{2} \left( \frac{5d'}{16B} \right)^2 \right) \leq \sqrt{B}\eta\alpha \frac{45d'^2}{2 \cdot (16B)^2}.$$

This concludes the proof that $O_S$ is well defined. To prove we never leave the unit ball if $\mathcal{E}$ as depicted in Equation (4) holds, notice that this is exactly the event where we either output 0 or execute one of Items 2a to 2c in the oracle. We will show by induction that for all $t \in [T]$:

$$\|w_{t+1}\|^2 = \|w_t - \eta O_S(S_{1:t}, w_t, V_t)\|^2 \leq 2\eta^2\alpha^2(t+1).$$

For the base case $\|w_0\| = 0 \leq 2\eta^2\alpha^2$. Now assume it holds for $t$ and we will prove for $t+1$. Consider the case where the first type of update is performed with $w_t(I_j) = w_t(I_{j+1})$:

$$
\begin{aligned}
\|w_{t+1}\|^2 &= \|w_t - \eta\alpha e_{I_{j+1}} + \eta\alpha e_{I_j}\|^2 \\
&= \sum_{i=1}^{j-1}(w_t(I_s))^2 + (w_t(I_j) + \eta\alpha)^2 + (w_t(I_{j+1}) - \eta\alpha)^2 + \sum_{s=j+2}^{\frac{7d'}{16B}}(w_t(I_s))^2 \\
&= \sum_{s=1}^{\frac{7d'}{16B}}(w_t(I_s))^2 + 2\eta\alpha(w_t(I_j) - w_t(I_{j+1})) + 2\eta^2\alpha^2 \\
&= \sum_{i=1}^{\frac{7d'}{16B}}(w_t(I_s))^2 + 2\eta^2\alpha^2 \\
&\leq 2\eta^2\alpha^2 \cdot t + \eta^2\alpha^2 = \eta^2\alpha^2(1+t).
\end{aligned}
$$

And if the second type of update occurs:

$$
\begin{aligned}
\|w_{t+1}\|^2 &= \|w_t + \eta\alpha e_{I_j}\|^2 \\
&= \sum_{s\neq j}(w_t(I_s))^2 + \eta^2\alpha^2 \\
&\leq 2\eta^2\alpha^2 t + \eta^2\alpha^2 \leq 2\eta^2\alpha^2.
\end{aligned}
$$

Since $\alpha = \min\left\{1, \frac{1}{\eta\sqrt{2T}}\right\}$, this shows that for all $t \in [T]$: $\|w_t\| \leq 2\eta^2\alpha^2 T = 2\eta^2 T \cdot \left\{1, \frac{1}{2\eta^2 T}\right\} \leq 1$. $\qquad\square$

## C. Proof of Theorem 4.1

From the reduction in Lemma D.2 it suffices to prove the result for a sample-dependent oracle as defined in Appendix D. This is established in the following Lemma:

**Lemma C.1.** *For every $n \geq 17$, $d = 712n\log n$ and $\eta > 0$, there are a finite datapoint set $Z$ and a distribution $\mathcal{Z}$ over $Z$, a 3-Lipschitz convex function $f(w, z)$ in $\mathbb{R}^{2d}$ and sample-dependent gradient oracle $O_S$ such that with probability $\frac{1}{2}$ if we run one-pass SGD with $\eta$ as a learning rate for $n$ steps then for every suffix averaging $\tau \in [n+1]$:*

$$
F(\widehat{w}_{n,\tau}) - F(0) = \Omega\left(\min\left\{\eta\sqrt{n}, 1\right\}\right).
$$

*Proof of Lemma C.1.* Since $d \geq 256$, from Lemma 1 in Schliserman et al. (2024), there exists $U \subset \left\{\frac{1}{\sqrt{d}}, -\frac{1}{\sqrt{d}}\right\}^d$ such that $|U| \geq 2^{\frac{d}{178}} \geq 2^n$ and for all $u \neq v \in U$: $|\langle u, v\rangle| \leq \frac{1}{8}$. This implies that:

$$
\forall u \neq v \in U : |\{i \in [d] : u(i) = v(i)\}| \leq \frac{9}{16}d. \tag{11}
$$

Let $Z = P(U)$, and let $\mathcal{Z}$ be a distribution over $Z$ such just for a new sample $V \sim \mathcal{Z}$ every $u \in U$ will be in $V$ with probability $\delta = \frac{1}{4n^2}$. Let $\alpha = \min\{1, \frac{1}{\eta\sqrt{n}}\}$. Let $W$ $\{x \in \mathbb{R}^{2d} : \|x\| \leq 1\}$. Denote $w^{(1)} = w[1:d], w^{(2)} = w[d+1:2d]$. For a subset of indices $I \subset [d]$ denote the following:

$$
e(I) = \frac{1}{|I|}\sum_{i\in I}e_i \quad \text{and} \quad w(I) = \frac{1}{|I|}\sum_{i\in I}w(i).
$$

We will consider the following function with blocks of indices of size at most $8\log n$:

$$
f(w, V) = \max_{v\in V}\left\{\frac{9\alpha}{16\cdot 2\sqrt{2}}\eta\sqrt{n}, \left(w^{(1)} + w^{(2)}\right)\cdot v\right\} \tag{12}
$$
$$
+ \max_{I\subset d:|I|\leq 8\log n}\left\{0, w^{(1)}(I)\right\} + \max_{I\subset d:|I|\leq 8\log n}\left\{0, -w^{(2)}(I)\right\}.
$$

It holds that $f$ is convex and 3-Lipschitz. Now assume we have an order over the vectors in $U$ which is lexicographic. Denote by $u_{(r)}$ the $r$-th vector in such an order. We will be interested in the following event:

$$\mathcal{E} = \left\{ \cap_{i=1}^{\lceil \frac{n}{16} \rceil} V_i \neq \emptyset \text{ and } \min_{r \in [|U|]} \{u_{(r)} : u_{(r)} \in \cap_{i=1}^{\lceil \frac{n}{16} \rceil} V_i\} \in \cap_{i=\lceil \frac{n}{16}+1 \rceil}^{n} \overline{V_i} \right\}. \tag{13}$$

From Lemma E.3, $\mathcal{E}$ occurs with probability at least $\frac{1}{2}$. From now on we will assume $\mathcal{E}$ occurs. Let $S = \{V_1, \ldots, V_n\}$ be some training set. Next we define a sample dependent oracle $O_S$. Given $w_t$ and at all past samples $S_{1:t} = \{V_1, \ldots, V_t\}$:

1. If $|S_{1:t}| \leq \lceil \frac{n}{16} \rceil$ output 0.

2. Otherwise check if the exists $u_0 \in \cap_{i=1}^{n} V_i$. We will check this in lexicographic order over vectors in $U$, so we will find the minimal such vector.

   (a) If $\cap_{i=1}^{\lceil \frac{n}{16} \rceil} V_i = \emptyset$ or $u_0 \in \cup_{t'=\lceil \frac{n}{16} \rceil+1}^{t} V_{i_{t'}}$, output an arbitrary sub-gradient.
   (b) Otherwise compute the following sets:

$$J_t^p = \left\{ 2i : i \in [d] \text{ and } u_0(i) > 0 \text{ and } \left[w_t^{(1)} + w_t^{(2)}\right](i) = 0 \right\}$$

$$J_t^n = \left\{ i \in [d] : u_0(i) < 0 \text{ and } \left[w_t^{(1)} + w_t^{(2)}\right](i) = 0 \right\}.$$

   We will use $w^{(1)}$ to take steps in the negative coordinates of $u_0$ - $J_t^n$ - and $w^{(2)}$ to take steps in the positive coordinates of $u_0$ - $j_t^p$. Choose $\min\{8\log n, |J_t^p|\}$ indices out of $J_t^p$ denote them $P_t$ and $\min\{8\log n, |J_t^n|\}$ out of $J_t^n$ denote them $N_t$. Output

$$\alpha \left(e(N_t) - e(P_t)\right).$$

We will prove this oracle is valid and when $\mathcal{E}$ holds not projections take place. Its proof is deferred to Appendix C.1.

**Lemma C.2.** $O_S$ *stated above is a valid sample dependent first order oracle of $f$ defined in Equation* (12). *Furthermore if $\mathcal{E}$ holds it will induce a trajectory such that we never leave the unit ball and no projections take place.*

Assuming $\mathcal{E}$ occurs, for $T' = \lceil \frac{n}{16} \rceil + \lceil \frac{d}{8\log n} \rceil \leq \frac{3n}{16} + 2$ we have that

$$\forall t \geq T' : \left[w_t^{(1)} + w_t^{(2)}\right] = \frac{\alpha\eta\sqrt{d}}{\sqrt{8\log n}} u_0 = \frac{\alpha\eta\sqrt{n}}{2\sqrt{2}}.$$

So for every suffix averaging $\tau$:

$$\begin{aligned}
\left[\widehat{w}_{n,\tau}^{(1)} + \widehat{w}_{n,\tau}^{(2)}\right] \cdot u_0 &\geq \left(1 - \frac{T'}{n}\right) \left[w_T^{(1)} + w_{T'}^{(2)}\right] \cdot u_0 \\
&= \left(1 - \frac{\frac{3n}{16}+2}{n}\right) \cdot \frac{\alpha\eta\sqrt{n}}{2\sqrt{2}} \|u_0\|^2 \\
&\geq \left(1 - \frac{\frac{3n}{16}+\frac{n}{8}}{n}\right) \cdot \frac{\alpha\eta\sqrt{n}}{2\sqrt{2}} \qquad\qquad n \geq 16 \\
&= \frac{11}{16} \cdot \frac{\alpha\eta\sqrt{n}}{2\sqrt{2}}.
\end{aligned}$$

We now have that with probability $\frac{1}{2}$:

$$F_S(w_{n,\tau}) - F_S(0) \geq \frac{1}{n}\left(\frac{n}{16} \cdot \frac{11\alpha\eta\sqrt{n}}{16 \cdot 2\sqrt{2}} + \frac{7n}{8} \cdot \frac{9\alpha\eta\sqrt{n}}{16 \cdot 2\sqrt{2}}\right) - \frac{9\alpha\eta\sqrt{n}}{16 \cdot 2\sqrt{2}} \geq \frac{\alpha\eta\sqrt{n}}{365} = \frac{1}{365} \cdot \min\{1, \eta\sqrt{n}\}.$$

$\square$

### C.1. Proofs for Auxiliary Lemmas

*Proof of Lemma C.2.* For the first part of the oracle note that $0 \in \partial f(0, V)$ so we can stay at 0 for as long as we like, in particular for $\lceil \frac{n}{16} \rceil$ steps. For the second part if $\cap_{i=1}^{\lceil \frac{n}{16} \rceil} V_i = \emptyset$ we keep outputting 0 which is allowed as we saw before. If $\cap_{i=1}^{\lceil \frac{n}{16} \rceil} V_i \neq \emptyset$ we choose some $u_0$. If $u_0 \in \cup_{i=\lceil \frac{n}{16} \rceil + 1}^{t} V_i$ we output a valid sub gradient by definition, if it is not we are clearly a taking gradient step for $\alpha h(w)$. It is thus left to show that in this case $0 \in \partial g(w_t, V)$. Indeed, in this event until this point the gradient only output 0 or executed Item 2b in the oracle so the nonzero coordinates in $w_t$ have the same sign as $u_0$. Also for all $v \in V_{i_t}$ we have that $v \neq u_0$ which suggests using Equation (11):

$$\left(w_t^{(1)} + w_t^{(2)}\right) \cdot v \leq \sum_{i:v(i)=u_0(i)} \left(w_t^{(1)}(i) + w_t^{(2)}(i)\right) \cdot v(i) \leq \frac{\alpha\eta}{\sqrt{8\log n}}\sqrt{d} \cdot \sum_{i:v(i)=u_0(i)} u_0 \cdot v(i) \leq \frac{9\alpha\eta}{16 \cdot 2\sqrt{2}}\sqrt{n}.$$

This concludes the proof that $O_S$ is well defined. To see no projections take place if $\mathcal{E}$ as depicted in Equation (13) holds, note that this is exactly the event where the oracle either outputs 0 or executes Item 2b in the oracle. In this event:

$$\forall t \in [n], i \in [d] : |w_t(i)| \leq \frac{\alpha\eta}{\sqrt{8\log n}} \implies \|w_t\|^2 \leq \frac{\alpha\eta\sqrt{2d}}{\sqrt{8\log n}} = \frac{1}{2}\min\{\eta\sqrt{n}, 1\} \leq \frac{1}{2}.$$

$\square$

## D. Reduction to Sample-Dependent Oracle

For the reduction we will further formalize the notion of sample-dependent oracle when a gradient oracle $O_z$ is data-dependent and at step $t$ it is allowed to depend on the examples seen up to this step. We denote the sample-dependent oracle by $O_S$ when $S = (S_1, \ldots, S_T)$ and for $t \in [T]$, $S_t$ is the set of examples given to the algorithm at step $t$. For example, for SGD $S_t = \{z_{i_t}\}$ and for Gradient Descent $S_t = S$. Then we denote the following:

$$O_S(S_{1:t-1}; S_t, w_t) = \frac{1}{|S_t|} \sum_{z \in S_t} O_z(S_{1:t-1}; w) \quad \text{when } O_z(S_{1:t-1}; w) \in \partial f(w, z).$$

when $S_{1:0} = \emptyset$, $S_{1:t} = (S_1, \ldots, S_t)$. Finally we denote the trajectory induced by $O_S$ which is initialized at $w_0 = 0$ and is specified by the following equation:

$$w_{t+1}^S = w_t^S - \eta O_S(S_{1:t-1}; w_t, S_t). \tag{14}$$

The following Lemma from Livni (2024) will prove the reduction to the sample dependent oracle case:

**Lemma D.1** (Lemma 9 in Livni (2024)). *Suppose $q \in \mathbb{R}^T$, $\|q\|_\infty \leq 1$ and $Z$ is finite. And suppose that $f(w, z)$ is a convex, $L$-Lipschitz function over $w \in \mathbb{R}^d$, let $\eta > 0$, let $O_S$ be a sample dependent first order oracle, and for every sequence of samples $S = (S_1, \ldots, S_T)$ define the sequence $\{w_t^S\}_{t=1}^{T+1}$ as in Equation (14).*

*Then, for every $\epsilon > 0$ there exists an $L + 1$-Lipschitz convex function $\bar{f}((w, x), z)$ over $\mathbb{R}^{d+1}$ (that depends on $q, f, T, \eta, n, O_S, \epsilon$) such that for any oracle $O_z$ for $\bar{f}(z, \cdot)$, define $u_0 = 0 \in \mathbb{R}^d$ and $x_0 = 0 \in \mathbb{R}$ and*

$$(u_t, x_t) = (u_{t-1}, x_{t-1}) - \frac{\eta}{|S_t|} \sum_{z \in S_t} O_z((u_t, x_t))$$

*then if we define*

$$u_q = \sum_{t=1}^{T} q(t)u_t \quad x_q = \sum_{t=1}^{T} q(t)x_t \quad w_q^S = \sum_{t=1}^{T} w_t^S$$

*we have that $u_q = w_q^S$ and for all $z$:*

$$|\bar{f}((u_q, x_q), z) - f(w_q^S, z)| \leq \epsilon$$

$$|\bar{f}((0, 0), z) - f(0, z)| \leq \epsilon.$$

The following Lemma easily follows:

**Lemma D.2.** *Suppose $Z$ is finite, $f(w, z)$ is a convex, $L$-Lipschitz function over $w \in \mathbb{R}^d$, let $\eta > 0$, let $O_S$ be a sample dependent first order oracle, and for every sequence of samples $S = (S_1, \dots, S_T)$ define the sequence $\{w_t^S\}_{t=1}^{T+1}$ as in Equation (14). Suppose also that for some number of steps $T$ and $\tau \in [T+1]$ we have with probability $p$:*

$$F(\widehat{w}_{T,\tau}^S) - F(0) \geq \ell$$

*Where $\ell > 0$. Then there exists an $L+1$-Lipschitz convex function $\bar{f}((w, x), z)$ over $\mathbb{R}^{d+1}$ (that depends on $\tau, f, T, \eta, n, O_S, \epsilon$) such that for any oracle $O_z$ for $\bar{f}(z, \cdot)$ if we define $v_0 = 0 \in \mathbb{R}^{d+1}$ and*

$$v_{t+1} = v_t - \frac{\eta}{|S_t|} \sum_{z \in S_t} O_z(v_t),$$

*we will have:*

$$\bar{F}(\widehat{v}_{T,\tau}) - \bar{F}(0) \geq \frac{\ell}{2}.$$

*Proof.* Let $\epsilon = \frac{\ell}{4} > 0$, define $q \in \mathbb{R}^d$ as follows:

$$q(t) = \begin{cases} \frac{1}{T-\tau+2} & \tau \leq t \leq T+1 \\ 0 & \text{otherwise} \end{cases}.$$

For this $q, \epsilon$ let $\bar{f}$ be the function whose existence follows from Lemma D.1. It is easy to see that $\widehat{w}_\tau^S = w_q^S$ and $\widehat{v}_\tau = (u_q, x_q)$. Then with probability $p$ we have:

$$\bar{F}(\widehat{v}_{T,\tau}) - \bar{F}(0) = \bar{F}(u_q, x_q) - \bar{F}(0) \geq F(w_q^S) - F(0) - 2\epsilon \geq \ell - 2 \cdot \frac{\ell}{4} = \frac{\ell}{2}.$$

$\square$

# E. Additional Lemmas

**Lemma E.1** (Lemma 14 in Koren et al. (2022)). *For any step-size $\eta > 0$, $T \in \mathbb{N}$ and $d = \lceil 16\eta^2 T^2 \rceil$ there exists a deterministic convex optimization problem $h : W \to \mathbb{R}$ where $W \subset \mathbb{R}^{d+1}$ is of constant diameter such that:*

$$h(\widehat{w}) - \min_{w \in W} h(w) \geq \frac{1}{8} \min\left\{\frac{1}{\eta T} + \eta, 1\right\}.$$

**Lemma E.2.** *Let $U$ be a subspace such that $|U| \geq 2^n$. Let $Z = P(U)$ and let $\mathcal{Z}$ be a distribution of $Z$ such that for a random sample $V \sim \mathcal{Z}$ every $u \in U$ is in $V$ with probability $\frac{1}{2}$. Then for a sample $S = \{V_1, \dots, V_n\} \sim \mathcal{Z}^n$ with probability at least $\frac{1}{2}$ it holds that: $\cup_{i=1}^n V_i \neq U$*

*Proof.* First it holds that:
$$\Pr\left[u \in \cup_{i=1}^n V_i\right] = 1 - \Pr\left[u \notin \cup_{i=1}^n V_i\right] = 1 - 2^{-n}.$$

This implies:

$$\Pr\left[\cup_{i=1}^d V_i = U\right] = \Pr\left[\forall u \in U, u \in \cup_{i=1}^n V_i\right] = (1 - 2^{-n})^{|U|} \leq (1 - 2^{-n})^{2^n} \leq \frac{1}{e} < \frac{1}{2}.$$

$\square$

**Lemma E.3** (Lemma 9 in Schliserman et al. (2024)). *For $d = 712n \log n$ and $U_d$ as depicted in Lemma 1 from Schliserman et al. (2024), let $Z = P(U)$ and let a distribution $\mathcal{Z}$ over $Z$ be such that for $V \sim \mathcal{Z}$ every $u \in U_{d'}$ is in $V$ with probability $\frac{1}{4n^2}$. Let $S = \{V_1, \dots, V_n\}$ be a training set drawn i.i.d from $\mathcal{Z}$. Denote $P_t = \cap_{i=1}^{t-1} V_i$ and $S_t = \cap_{i=t}^n \overline{V}_i$. More over if $P_t \neq \emptyset$ we denote according to some ordering $i \mapsto v_i$ of the vectors in $U$, $r_t = \arg\min\{r : v_r \in P_t\}$, and $J_t = v_{r_t} \in U_d$, denote the following event:*

$$\mathcal{E} = \{\forall t \leq T, P_t \neq \emptyset \text{ and } J_t \in S_t\}.$$

*Then if $T = n$, $\Pr[\mathcal{E}] \geq \frac{1}{2}$. In particular with probability at least $\frac{1}{2}$:*

$$P_{\lceil n/16 \rceil} \neq \emptyset \text{ and } J_{\lceil n/16 \rceil} \in S_{\lceil n/16 \rceil}.$$

.

