# OpenReview forum: "Rapid Overfitting of Multi-Pass SGD in Stochastic Convex Optimization"
_ICML.cc/2025/Conference — ICML 2025 spotlightposter_

### Official Review · Reviewer_4BeX · 2025-03-12

**Overall Recommendation:** 4

**Summary:**

The author(s) analyzed the generalization lower bound of SCO in the multi-pass scenario. The lower bound that SCO with multi-pass can quickly overfit and yield $\Theta(1)$ population loss.

**Claims And Evidence:**

Yes

**Essential References Not Discussed:**

No

**Experimental Designs Or Analyses:**

This is a pure theoretical work and there is no numerical experiments.

**Methods And Evaluation Criteria:**

Yes

**Other Comments Or Suggestions:**

right column of line 255, $\partial h_1(0) 0 $, is this a typo?

**Other Strengths And Weaknesses:**

Pros:
- The writing of the paper is crispy, it is a joy to read the paper;
- The lower bound of SCO in Theorem 3.1 and 3.2 are new and interesting to me. The analysis is also neat.

Cons:
- The theoretical result is some kind of inconsistent with practitioner's observation as multi-pass SGD usually doesn't really hurt the generalization. Better to add a limitation section to address this issue.

**Questions For Authors:**

Merely looking at Theorem 3.1, setting $\eta = T^{ -3/4 }$ yields population risk converges to 0. Why can't we do this?

**Relation To Broader Scientific Literature:**

No

**Theoretical Claims:**

The proofs looks reasonable to me. I did not check the proof in detail.

---

> ### Author Rebuttal · Authors · 2025-03-31
>
> Thank you for the thorough review and input. Below are our responses to the main comments in your review:
>
> > “The theoretical result is some kind of inconsistent with practitioner's observation as multi-pass SGD usually doesn't really hurt the generalization. Better to add a limitation section to address this issue.”
>
> Thanks for this comment. We will add a discussion addressing this limitation. Needless to say, lower bounds depict the worst case and not “typical” cases, which are arguably those encountered in practice. In fact, even in theory, if we add further assumptions such as strong convexity, then multiple passes will no longer hurt performance. We will add this comparison to practical observations, as well as related theoretical setups, in the final version.
>
> > ”Merely looking at Theorem 3.1, setting $\eta = T^{ -3/4 }$ yields population risk converges to 0. Why can't we do this?”
>
> You are correct: the lower bound is matched by Bassily’s stability upper bound (up to an additive term of $\eta T/n$). So setting $\eta = T^{-3/4}$ will achieve $T^{-1/4} = 1/ ({nK})^{1/4}$ when $n$ is the sample size and $K$ is the number of epochs. Noticeably, in order to achieve the optimal $1/\sqrt{n}$ rate one would need as many as $K = n$ epochs (i.e., $T=n^2$ total number of steps). We will discuss these observations in the final version.

---

### Official Review · Reviewer_M6q6 · 2025-03-12

**Overall Recommendation:** 3

**Summary:**

This paper considers multi-pass SGD in the SCO setting. Single pass SGD is known to achieve optimal excess population error, but it was not clear how it performs in terms of population loss after multiple passes. In fact, this paper shows lower bounds indicating that (several versions of) multi-pass can quickly "overfit" after even 2 passes over the training data and can have Omega(1) excess population risk when using the single-pass-optimal O(1/sqrt{n}) stepsize.

**Claims And Evidence:**

The claims in the paper are supported by proofs. However, I would not necessarily describe these proofs as "clear". This can probably be partially attributed to the fact that the underlying problem and arguments are quite complicated and delicate, but I found it difficult to follow the details of the proofs in the paper.

The proof sketches give some idea of what is going on, but a key part of the argument rests on Livni's 2024 connection between "sample dependent oracles" and a standard SCO oracle. This connection is hardly explained at all, and so a thorough understanding of this paper appears to depend on the reader already being intimately familiar with Livni 2024 (I am not). Perhaps this is unavoidable, but I think the paper could be improved and made more readable by recapping this paper's Lemma D.1 from Livni 2024 and explaining what is going on there.

In the same vein, the proof of Lemma 5.1 is very terse and difficult to follow in full detail. It appears that an effort was made to cram it into the page limit. I would argue for making it easier to follow, even if it requires punting some/all of it to the appendix.

**Essential References Not Discussed:**

Nothing comes to mind.

**Experimental Designs Or Analyses:**

N/A

**Methods And Evaluation Criteria:**

Yes, this is a theory paper and the implicit evaluation metric is, appropriately, the quality of the results of the theorems.

**Other Comments Or Suggestions:**

pg 5: "furher", "minimzer"

Section 3: the first sentence says "for K \leq n epochs", but Theorem 3.1 says "3 \leq K \leq n^2", so there is a discrepancy there.

Section 4: first sentence says "\Omega(\eta \sqrt{T})" but Theorem 4.1 says "\eta \sqrt{n}". This is consistent because it is referring to one pass SGD, but I would suggest using the same letter in both places.

**Other Strengths And Weaknesses:**

See previous comments. Generally, I think the paper is interesting and the prose are well-written and understandable, but I the proofs are quite terse and hard to follow--I think the paper would be greatly improved by making the technical ideas more accessible and readable.

**Questions For Authors:**

How much of this hinges on using a fixed stepsize? For instance, the h_2 portion of the argument in the Theorem 3.1 sketch seems like it could be complicated even by using variable stepsizes, even e.g., \eta + 10^{-100} N(0,1), because then the argmaxs would be a.s. unique after 2d steps so you could lose control over the iterates. The argument also seems to hinge similarly on initialization at zero (or else argmaxs might always be unique).  Is that correct or am I missing something? Do you see a path towards a similar lower bound for variable stepsizes or e.g. random initialization?

**Relation To Broader Scientific Literature:**

This is a very interesting paper that ties in well to the existing literature. Previously, it was mostly assumed that (1) one-pass SGD is optimal in terms of the excess population risk, (2) "few"-pass SGD is probably about the same or maybe even a little better in terms of excess population risk, and (3) that "many"-pass SGD might eventually overfit. This paper is a useful and thought provoking corrective to point (2), showing that even 2-3 passes over the training data are enough to overfit, or at least have substantially slower convergence (if the stepsize is chosen optimally, you get T^{-1/4} convergence for multi-pass instead of T^{-1/2} for one-pass).

**Theoretical Claims:**

Even though I would describe myself as an expert on lower bounds for convex optimization, I found it very difficult to follow these proofs, and I am nowhere close to 100% confident that I would have caught an error if there was one. That said, I did read through all of the proofs and did my best to follow them, and I found nothing that looked incorrect.

I will also add that these results hinge crucially on results from Livni 2024, which I was not previously familiar with and which I did not review in detail.

---

> ### Author Rebuttal · Authors · 2025-03-31
>
> Thank you for the thorough review and feedback. We respond to main claims below:
>
> > “a key part of the argument rests on Livni's 2024 connection between 'sample dependent oracles' and a standard SCO oracle. This connection is hardly explained at all”
>
> We thank you for the comment - we will improve this discussion for the final version. Notice that we do discuss it briefly in the main text (see left column line 290) and that we did add a section about the reduction done in Lemma D.1. But, we will add further intuition in the main text to make the result more accessible.
>
> > “In the same vein, the proof of Lemma 5.1 is very terse and difficult to follow in full detail. It appears that an effort was made to cram it into the page limit.” I would argue for making it easier to follow, even if it requires punting some/all of it to the appendix.”
>
> We will follow the reviewer’s advice and we will make it easier to follow, by deferring it to the appendix if needed.
>
> > ”How much of this hinges on using a fixed stepsize? For instance, the h_2 portion of the argument in the Theorem 3.1 sketch seems like it could be complicated even by using variable stepsizes [...] The argument also seems to hinge similarly on initialization at zero [...]  Do you see a path towards a similar lower bound for variable stepsizes or e.g. random initialization?”
>
> These are great questions, indeed certain variants of SGD pose a challenge to our proof technique, and it is important to add a discussion to the paper on that. For some variants of SGD such as dynamic learning rate, our technique can be utilized, perhaps with a slight increase in the dimension (when using $h_1$ and not $h_2$, see right column line 251).
>
>
> As for random initialization - you are correct that our construction relies on deterministic initialization. Showing lower bounds for random initialization is an interesting open problem, and in fact not just in the context of multi-pass SGD but also for full batch GD, for example.

---

### Official Review · Reviewer_UpAF · 2025-03-14

**Overall Recommendation:** 2

**Summary:**

This work considers the Stochastic Convex Optimization (SCO) setting and investigates the excess population risk and sample complexity lower bounds for Stochastic Gradient Descent (SGD). While the majority of previous work tackled GD or single-pass SGD, this paper mainly focuses on the multi-pass version of SGD. More specifically, the authors derive excess population risk lower bounds of $\Omega(\eta\sqrt{T} + 1/\eta T)$ for several versions of multi-pass SGD, both with and without replacement, when the number of passes exceeds a small constant. These lower bounds are tight and match the corresponding upper bounds. Additionally, the authors present a novel empirical risk lower bound of $\Omega(\sqrt{\eta}n)$ for single-pass SGD, further enriching the understanding of SGD's generalization properties.

**Claims And Evidence:**

Not applicable.

**Essential References Not Discussed:**

The authors appropriately cite the most relevant prior work and provide a clear and detailed discussion of how their contributions relate to and advance the existing literature.

**Experimental Designs Or Analyses:**

Not applicable.

**Methods And Evaluation Criteria:**

Not applicable.

**Other Comments Or Suggestions:**

Not applicable.

**Other Strengths And Weaknesses:**

**Strengths**

Generally speaking, the paper is well-written, with a clear exposition that makes it easy to follow. The proof sketch is intuitive and highlights the key ingredients and techniques used in constructing lower bounds. From a contribution perspective, this paper gives a nearly complete picture for the understanding of excess population risk lower bound of multi-pass SGD under the non-smooth SCO setting. These results offer novel insights into the generalization property of multi-pass SGD and the relationship between overfitting, step-size and number of passes.

**Weaknesses**

However, the major weakness of this paper lies in its limited novelty. It is evident that the excess risk lower bound in Theorem 3.1 and Theorem 3.2 share an identical form with those in [Amir et al., 2021] and [Bassily et al., 2020]. Furthermore, as also acknowledged by the authors, the technique used to construct these lower bounds are direct extension of the technique used in [Amir et al., 2021]. While the results in this paper are technically new and are different from prior work such as [Amir et al., 2021], which tackles only GD, and [Bassily et al., 2020], which provides only uniform stability lower bounds, the conclusions in this paper are not entirely surprising and unexpected. Given this lack of significant improvement, I tend to reject this work unless the authors could provide more compelling arguments to address these concerns or offer deeper insights into the open questions discussed in the paper.

**Questions For Authors:**

No further questions.

**Relation To Broader Scientific Literature:**

This work is entirely theoretical and does not present any negative broader scientific or societal impacts.

**Theoretical Claims:**

The major theoretical conclusions are reasonable, and the key steps in the proofs appear to be correct.

---

> ### Author Rebuttal · Authors · 2025-03-31
>
> Thank you for the detailed review and discussion. We respond to the main claims below:
>
> > It is evident that the excess risk lower bound in Theorem 3.1 and Theorem 3.2 share an identical form with those in [Amir et al., 2021] and [Bassily et al., 2020]. Furthermore, as also acknowledged by the authors, the technique used to construct these lower bounds are direct extension of the technique used in [Amir et al., 2021]. [...] the conclusions in this paper are not entirely surprising and unexpected.”
>
> We see that a more thorough and in-depth comparison of our work to these two references is warranted, and we will make this improvement in the paper. However, the lower bounds in 3.1 and 3.2 do **not** share an identical form with Amir et al. nor with Bassily et al, and the technique used to construct the lower bounds are **not** direct extensions of Amir et al.
>
> *Regarding Bassily et al:*
>
> As you point out, Bassily et al shows a lower bound on stability and not for the generalization gap. One could argue that bad generalization in the face of instability is not surprising. But then at the same time SGD (single pass) counters this statement: it is provably a non-stable algorithm that achieves optimal generalization performance.
>
> We also note that Bassily et al’s lower bound for stability applies to all epochs of SGD, whereas SGD learns at the first epoch and overfits from the second epoch (as we show). It is unclear how one could derive an intuition for such a behavior from the results of Bassily et al.
>
> Overall our proof indeed relies on generating instability (which is a necessary condition for overfitting), but is more involved and incorporates further techniques and constructions such as the usage of Feldman’s function, as well as the notion of sample-dependent oracle which we modify and strengthen to fit our setup.
>
> *Regarding Amir et al:*
>
> One major challenge we faced, given Amir et al’s work, is that we aimed for constructions in dimension linear in the sample-size. This differs greatly from Amir et al that exhibited an exponential dependence of the dimension on the sample size, and did pose several challenges: for example, we require working with Feldman’s function which is a more subtle construction than previous existing ones in high (exponential) dimension.
>
> Some of these challenges have been tackled before in the context of GD (as opposed to multi-pass SGD), and we incorporate some of these ideas. But our results deal with SGD which, unlike GD that was discussed in previous papers [Amir et. al., 2020, Livni, 2024], does generalize in its first epoch. In particular, it provably circumvents previous hardness constructions in its first epoch, and begins its second epoch from an optimal state in terms of generalization performance.
>
> Given this, we actually think it is a surprising result that SGD’s generalization rapidly deteriorates and has the same asymptotic bounds as its stability so quickly after the first pass (but not during the first pass).

---

### Official Review · Reviewer_E6PJ · 2025-03-19

**Overall Recommendation:** 3

**Summary:**

The paper makes three main contributions in Stochastic Convex Optimization with convex and Lipschitz (but not necessarily smooth) loss functions: First, they establish tight bounds on the population excess risk of multi-pass SGD that apply to both single-shuffle and multi-shuffle variants. Second, they prove similar tight bounds for with-replacement SGD that hold after a logarithmic number of steps. Finally, they provide new lower bounds on the empirical risk of single-pass SGD in nearly linear dimension, improving upon previous results that required quadratic dimension.

**Claims And Evidence:**

All claims are supported by mathematical proofs:

1. For the multi-pass SGD bounds:
- They provide a lower bound construction in Section 5 (Theorems 3.1 and 3.2)
- They prove matching upper bounds in Appendix A (Theorems 3.3 and 3.4)
- The proofs involve constructing specific loss functions and showing both that these functions are valid (convex, Lipschitz) and that SGD behaves as claimed when applied to them

2. For with-replacement SGD:
- They use similar techniques but add analysis involving the coupon collector's problem to handle the random sampling
- The proofs appear in Section 5 and Appendix A

3. For the improved empirical risk bounds:
- They provide a construction in nearly linear dimension in Section 4 (Theorem 4.1)
- The proof leverages techniques from previous works, but achieves better dimension dependency

Most proofs are constructive, and seemingly easy to follow given the proof outlines. I did not look into the appendix thoroughly.

**Essential References Not Discussed:**

None that I am aware of, though given the structure of many of the proofs they discuss it seems a bit surprising that discussions surrounding online convex optimization and adversaries were not discussed in more detail.

**Experimental Designs Or Analyses:**

None.

**Methods And Evaluation Criteria:**

None.

**Other Comments Or Suggestions:**

- There are a couple instances of spelling and grammar issues throughout the paper which can probably addressed by passing tex file through a spell checker.
- Might want to consider some restructuring, not having a conclusion to tie things together seems a bit odd.
- Is there any way to provide empirical verification of any of the theoretical claims to illustrate how relevant this is to practical optimization?

**Other Strengths And Weaknesses:**

Strengths:
1. Theoretical rigor - the paper provides complete proofs and careful mathematical constructions for all its claims
2. Novel insights - reveals a previously unknown phase transition between first and subsequent passes of SGD
3. Comprehensive analysis - covers multiple variants (with-replacement, single-shuffle, multi-shuffle)
4. Improves on prior work - achieves better dimension dependency than previous results
5. Clear practical implications - helps explain why multiple passes might lead to overfitting in the non-smooth setting

Weaknesses:
1. Theoretical focus - lacks empirical validation or real-world experiments to demonstrate the practical impact
2. Construction-based proofs - relies on specific constructed examples rather than showing results for general cases
3. Dimensionality assumptions - some results require high-dimensional settings that may not reflect practical scenarios
4. Limited practical guidance - doesn't provide clear recommendations for practitioners on how to avoid the identified overfitting issues

**Questions For Authors:**

Practical Implications:
- How do your theoretical results translate to practical guidance for practitioners?
- Are there specific strategies you'd recommend to mitigate the overfitting in multi-pass SGD?
- Have you observed the behavior your theory describes empirically in any real-world optimization problems?

Extension Possibilities:
- Could your analysis extend to other variants of SGD (e.g., with momentum, adaptive methods)?
- What about non-convex settings?
- Are there connections to implicit regularization in deep learning?

**Relation To Broader Scientific Literature:**

I have a background in convex optimization / online convex optimization and am thus aware of online-to-batch arguments and the like. However, I have not read or interacted with many of the core works that this paper is built off of and thus might not fully appreciate the the impact of this work in the broader scientific literature.

**Theoretical Claims:**

I did not verify fully the claims made by reviewing the appendix, however because most of the proofs are constructive none seemed incorrect at there face. That said, I am not an expert in this area, and thus some of the bounds discussed or the results derived may not actually make sense.

---

> ### Author Rebuttal · Authors · 2025-03-31
>
> Thank you for the thorough review and discussion. Following are our responses to your main comments:
>
> > “lacks empirical validation or real-world experiments to demonstrate the practical impact”
>
> Our main contributions are theoretical and, respectfully, we disagree that this is a weakness. Theoretical results, and specifically in Stochastic Convex Optimization, are well within the traditional scope of ICML and have significant impact on the broader study of machine learning.
>
> While theoretical results do not always offer immediate practical guidance, their goal is to improve understanding of SGD and they provide insight into the behavior of generalization in a simple setup (SCO). Such an understanding is a prerequisite for understanding more involved scenarios, such as those encountered in practical settings.
>
> > “Construction-based proofs - relies on specific constructed examples rather than showing results for general cases”
>
> Indeed, our main results are lower bounds --- and lower bounds are inherently construction based proofs. While lower bounds are demonstrated via specific constructions, they aid with providing a complete picture of the guarantees for the examined algorithms: whether we can find better algorithms, or that the algorithms we have at hand are already optimal.
>
> > “Dimensionality assumptions - some results require high-dimensional settings that may not reflect practical scenarios”
>
> The vast majority of our results hold already in dimension linear in sample size (and this is tight). Notably, the setting where the dimension is at least the sample size is currently the most interesting regime to study, in light of practical findings where overparameterized networks are shown to achieve superior performance. (In fact, in Theorem 4.1 for example, the main novelty is in reducing the dimension from quadratic in the sample size $n$, as in prior work, to linear in $n$.)
>
> The only exception to this is the case of Theorem 3.1 for $K=2$ epochs, which applies only in dimension exponential in $n$. However, for $K>2$ our proof construction is in dimension linear in $n$.
>
> >  “Limited practical guidance - doesn't provide clear recommendations for practitioners on how to avoid the identified overfitting issues”
>
> Indeed, we do not propose practical methods to avoid overfitting. Rather, we believe our contributions do give guidance to further theoretical studies, by indicating that in the general case (under the standard assumptions) multi-pass SGD quickly overfits, and additional assumptions will be needed in order to justify its empirical success.
>
> In particular, prior work indeed showed that overfitting can be circumvented by either avoiding overtraining (the classical online-to-batch argument) or through stabilizing the algorithm by decreasing the learning rate (c.f. stability arguments of Bassiley et al). This touches on an important aspect of our work, that shows that, in general, these two solutions are indeed exhaustive: overtraining without stability leads to overfitting.
>
> > “given the structure of many of the proofs they discuss it seems a bit surprising that discussions surrounding online convex optimization and adversaries were not discussed in more detail.”
>
> This is a very good point. In a way, our work demonstrates that the classic online-to-batch proof technique is a tight approach for establishing generalization of SGD (Theorem 3.1 shows that once the gradients are biased, like in epochs beyond the first, then generalization deteriorates; and Theorem 4.1 shows that stability/uniform convergence cannot be established in general). From a technical perspective however, we indeed do not invoke the online/regret analysis which is an established technique to achieve algorithmic upper bounds.
>
> We will add this discussion in the final version, thank you for this comment!
>
> > “Might want to consider some restructuring, not having a conclusion to tie things together seems a bit odd”
>
> The thing we had in mind is to have the discussion section appear at the end of the introduction (Section 1.2), and before the rest of the paper that delves into the technical proofs and formal statements. But, we will reconsider restructuring for the final version.
>
> > “Could your analysis extend to other variants of SGD (e.g., with momentum, adaptive methods)?”
>
> That is a very interesting question. Our results using the same techniques can be extended to SGD with varying stepsizes, perhaps with a slight increase in the dimension. As for other variants of SGD, it is currently unclear to us, and indeed an interesting question. We will add a discussion on further pursuing this avenue to the final version.
>
> > “What about non-convex settings?”
>
> Since most of the paper discusses lower bounds, the convex case subsumes the non-convex case, as the lower bounds are harder to achieve under the restriction of convexity (that is, any lower bound for the convex setting is also a lower bound for the non-convex setting).

---

### Decision · Program_Chairs · 2025-05-01

**Decision:**

Accept (spotlight poster)

**Comment:**

This paper derives tight bounds on the population excess risk of multi-pass SGD in the general non-smooth case, both with and without replacement. The results of this work connect well with the existing literature and offer new insights, in particular, that very few passes over the training data are already enough to overfit or at least have substantially slower convergence.

The paper also provides new lower bounds on the empirical risk of single-pass SGD in nearly linear dimension, improving upon previous results that required quadratic dimension.

One reviewer inquired about the technical differences between [Amir et al., 2021] and [Bassily et al., 2020]. The author clarified this in the rebuttal and will incorporate these comments into the final version of the paper.